# mRNA vaccine quality analysis using RNA sequencing

Helen M. Gunter[1,2,6], Senel Idrisoglu[1,2,6], Swati Singh[1,2], Dae Jong Han[2], Emily Ariens[2], Jonathan R. Peters[2], Ted Wong[3], Seth W. Cheetham[1,2], Jun Xu[4], Subash Kumar Rai[4], Robert Feldman[5], Andy Herbert[5], Esteban Marcellin[1], Romain Tropee[2], Trent Munro[1,2] & Tim R. Mercer[1,2] ✉

The success of mRNA vaccines has been realised, in part, by advances in manufacturing that enabled billions of doses to be produced at sufficient quality and safety. However, mRNA vaccines must be rigorously analysed to measure their integrity and detect contaminants that reduce their effectiveness and induce side-effects. Currently, mRNA vaccines and therapies are analysed using a range of time-consuming and costly methods. Here we describe a streamlined method to analyse mRNA vaccines and therapies using long-read nanopore sequencing. Compared to other industry-standard techniques, VAX-seq can comprehensively measure key mRNA vaccine quality attributes, including sequence, length, integrity, and purity. We also show how direct RNA sequencing can analyse mRNA chemistry, including the detection of nucleoside modifications. To support this approach, we provide supporting software to automatically report on mRNA and plasmid template quality and integrity. Given these advantages, we anticipate that RNA sequencing methods, such as VAX-seq, will become central to the development and manufacture of mRNA drugs.

mRNA vaccines were shown to be safe and effective during the COVID-19 pandemic[1,2] and many new mRNA treatments are being developed for a wide range of diseases, including other infectious pathogens, cancer, autoimmunity and cellular engineering. However, the effectiveness of these new treatments is dependent on the rapid and safe manufacture of mRNA at a sufficient scale, purity and integrity.

mRNA vaccines and therapies are made using rapid, cell-free in vitro transcription. mRNA manufacture begins with the preparation and linearisation of a plasmid DNA (pDNA) template that is transcribed in vitro by an RNA polymerase to generate a synthetic mRNA, with a 5′ cap and a template-encoded 3′ poly(A) tail typically incorporated (Fig. 1a)[3]. The mRNA must also be purified of contaminants, including antisense and double-stranded RNAs, that can elicit an innate immune

response that suppresses cellular translation and induces side-effects[4–11].

Rigorous analytics to measure mRNA quality are needed at each step in the manufacturing process. Undetected quality issues can cause reduced mRNA effectiveness, poor clinical trial outcomes and costly delays, and threaten regulatory approval. However, the analysis of mRNA vaccines is still evolving, and a range of different methods is currently needed to measure mRNA quality, including sequence identity, concentration, integrity, purity and safety[4,5,12]. Analysis of mRNA requires diverse techniques (such as RT-qPCR, Capillary and gel electrophoresis, RP-HPLC, IP-RP-HPLC and immunoblotting) that are onerous and expensive to maintain, and often cannot sensitively detect key mRNA quality features.

[1]Australian Institute for Bioengineering and Nanotechnology, University of Queensland, Brisbane, QLD, Australia. [2]BASE facility, University of Queensland, Brisbane, QLD, Australia. [3]Garvan Institute of Medical Research, Sydney, NSW, Australia. [4]Genome Innovation Hub, University of Queensland, Brisbane, QLD, Australia. [5]COVID19 Vaccine Corporation Limited (CVC), Auckland, New Zealand. [6]These authors contributed equally: Helen M. Gunter, Senel Idrisoglu. ✉e-mail: t.mercer@uq.edu.au

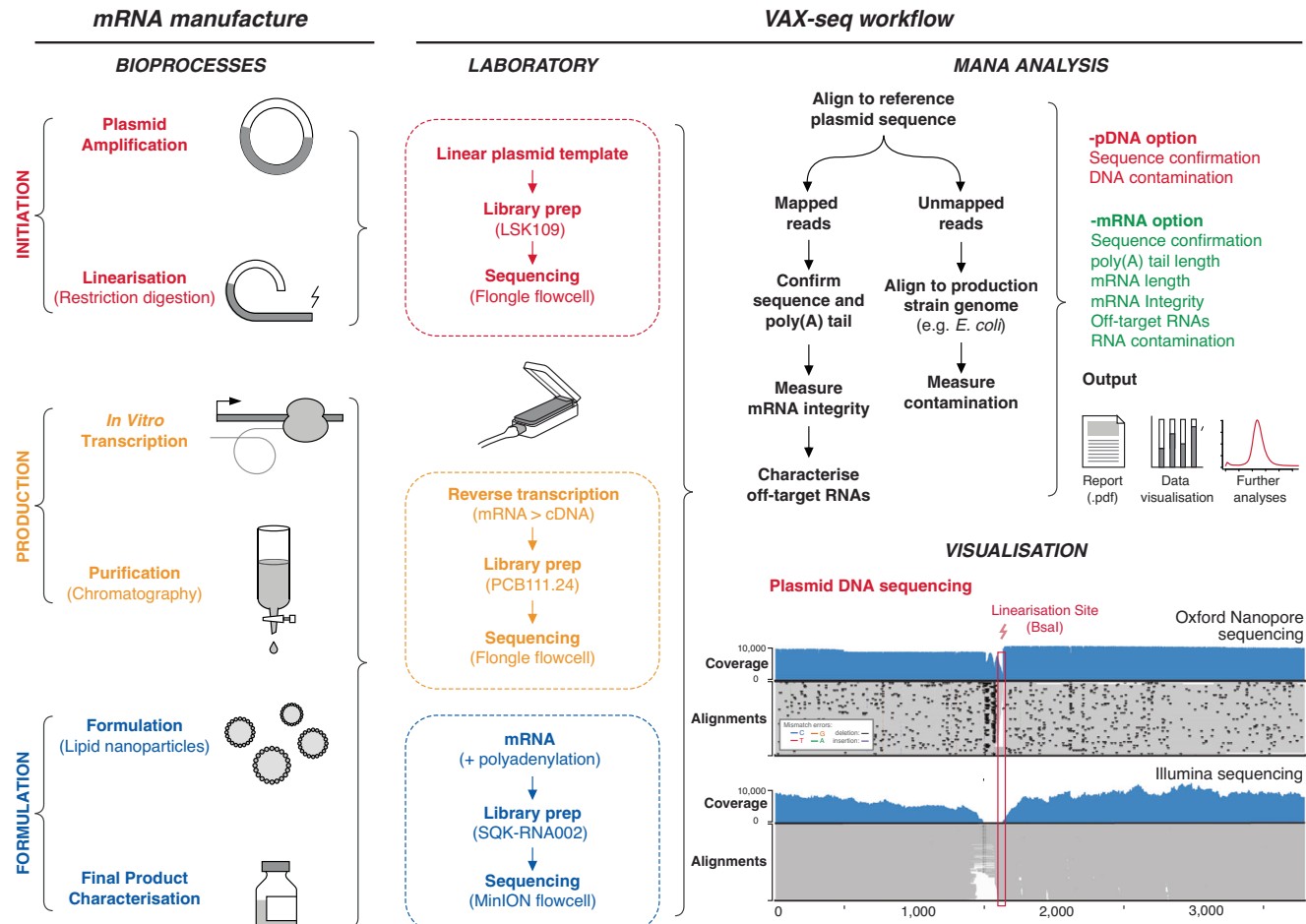

**Fig. 1 | mRNA vaccine production and VAX-seq workflow.** Schematic diagram illustrates the steps during mRNA manufacture (left panel), and the steps during VAX-seq analysis (right panel). This includes laboratory steps of long-read nanopore sequencing, followed by bioinformatic steps to analyse output data, including the supporting *Mana* software toolkit. mRNA vaccine quality features that can be analysed by VAX-seq are indicated (listed in red and green). In the bottom left corner is an IGV plot comparing Oxford Nanopore and Illumina sequencing of a plasmid DNA template. Coverage indicates the number of reads at each nucleotide position while the lower alignments grey bars indicate unique, individual alignments, with colouring indicating their similarity to the reference genome. Source data are provided as a Source data file.

RNA sequencing is widely used in the analysis of cellular gene expression, where it can determine the sequence identity and quantitative abundance of mRNAs. Additional RNA sequencing methods have been developed to measure polyadenylation, nucleoside modifications and RNA secondary structures[13–15]. However, RNA sequencing has not been used in the analysis and manufacture of mRNA drugs, despite the fact that they recapitulate the features of endogenous mRNAs. Here, we evaluate the use of both short- (Illumina (ILM)) and long-read (Oxford Nanopore Technologies (ONT)) sequencing for mRNA vaccine and therapy analysis.

Here we describe a streamlined protocol, termed VAX-seq, that uses long-read nanopore sequencing to measure the key quality features of mRNA vaccines and therapies. Unlike the diverse range of current industry-standard methods, VAX-seq comprises a single method, able to measure key mRNA quality attributes, including sequence identity, integrity, 3′-poly(A) tail length and DNA and RNA contamination. To support VAX-seq, we also develop a software toolkit, *Mana*, that provides detailed and automated reports of mRNA quality. We also show how the additional use of direct RNA sequencing can measure mRNA vaccine chemistry, including the incorporation of N1-methylpseudouridine[16].

## Results

### mRNA design and analysis with VAX-seq

Detailed analysis is key to the development and manufacture of mRNA vaccines and therapies. VAX-seq enables the analysis of mRNA vaccines using long-read nanopore cDNA sequencing (Fig. 1). To demonstrate the use and validation of the VAX-seq protocol, we designed and manufactured a reference eGFP mRNA.

The eGFP mRNA vaccine comprised the following components (in order from 5′ to 3′): (i) a T7 CleanCap promoter for in vitro transcription, (ii) an alpha-globin 5′ UTR, (iii) an enhanced green fluorescent protein (eGFP) open reading frame, (iv) an AES-mtRNR1 3′ UTR, (v) a 126nt poly(A) tail (with intervening 5nt linker sequence)[17], and (vi) a restriction enzyme (BsaI) digestion site for template linearisation (Fig. 4e). This construct was synthesised and cloned into a pUC-57 plasmid backbone (see 'Methods') and manufactured as described below. To support the analysis of VAX-seq data, we also developed a software tool, *Mana* (github.com/scchess/Mana) that receives aligned NGS libraries as input, and reports on plasmid and mRNA length, sequence identity and purity (Fig. S1). *Mana* can routinely generate a standardised report on mRNA samples and is suitable for documenting performance and final product characterisation.

## mRNA vaccine template preparation

The first mRNA manufacturing step is the preparation of the plasmid template, which is amplified in *E. coli*, extracted, purified, and linearised (see 'Methods'). Sequencing the plasmid template prior to in vitro transcription can identify mutations in the open reading frame and rearrangements that especially occur in low-complexity sequences such as the poly(A) tail. We sequenced the linearised pDNA template using short- (Illumina) and long-read (Oxford Nanopore) sequencing, with reads processed and aligned to the reference plasmid sequence.

We reliably confirmed the consensus accuracy of the plasmid sequence outside of the poly(A) tail region with long-read sequencing (Fig. S2a–c). VAX-seq can also measure the purity of linearised plasmid DNA template, and detect contaminants carried over from plasmid amplification. For example, in the eGFP mRNA we found the majority (86.8%) of reads aligned to the plasmid reference DNA, with the remaining unmapped reads derived from *E. coli* (6.8%) or failed reads (6.6%; Fig. S2d)[18]. Illumina short-read sequencing produced similar results, confirming sequence identity with high confidence and a slightly lower rate of contamination, which may have been due sequencing platform-specific differences (Fig. S3a–c).

During preparation, the supercoiled plasmid is cleaved using a restriction enzyme (BsaI) at the 3' end of the poly(A) tail to generate a linear template that prevents readthrough transcription (Fig. 4e). To measure the efficiency of plasmid linearization, HPLC, or agarose gel electrophoresis is typically used to distinguish linear from circular plasmids (Fig. S4a–d). Long-read nanopore sequencing can determine the full length of the linear plasmid, however ligation-based nanopore library preparation methods do not measure circular plasmids. We analysed the size of linearised plasmids using *Mana* (-plasmid option; Figs. 1 and S1), which provided a comparable size profile to industry-standard agarose gel and capillary electrophoresis methods (Fig. S4c–e).

## Analysis of mRNA sequence and structure

The linearised plasmid is then used as a template for in vitro transcription of the synthetic mRNA, which is purified of residual DNA, RNA and proteins and tested to confirm sequence identity, length and integrity[12,19]. We used long-read cDNA sequencing (SQK-PCS111) to analyse the purified mRNA, with reads processed and aligned to the reference plasmid sequence and analysed using *Mana* (-mrna option; Figs. 1 and S1; see 'Methods'). The consensus sequence identity and length of mRNA vaccines was easily confirmed (for individual reads, we measured a mean 5.0% error rate, with a higher deletion rate in the poly(A) tail (13%); Fig. 2a–c). A per-nucleotide comparison of the error profile between cDNA sequencing and previous plasmid DNA sequencing also reveals errors specific to the cDNA library preparation steps (Fig. S5a).

The poly(A) tail is needed for efficient translation of mRNA vaccines and is a key quality attribute to measure during manufacture[4,12,20]. VAX-seq anchors a reverse transcriptase primer to the 3' terminus of the poly(A) tail, enabling the complete poly(A) tail to be sequenced to measure its length (Figs. 2d and S6a, b). This measurement of poly(A) tail length from alignments showed a mean ~11.2% underestimation of length due to deletion errors (Figs. 2d and S2c). These errors were systematic, as they were reproducible between replicates (Fig. S5b). To normalise these deletion errors, we used *tailfindr* software to assess raw data and normalise the read-specific nucleotide translocation rate[21,22]. For the eGFP mRNA, we found *tailfindr* accurately estimated poly(A) tail length to be 126.04nt compared to the expected 126nt (Figs. S6b and 2e). This analysis was reproducible across replicate mRNA samples (Fig. S6c)[23].

For comparison, we also analysed synthetic mRNAs using short-read (Illumina) cDNA sequencing with the Illumina TruSeq mRNA Stranded Library Preparation Kit, with libraries analysed using an adapted *Mana* workflow (see 'Methods'). The consensus accuracy of short-read sequencing correctly confirmed mRNA sequence (Fig. 3a, b), however, misalignment of short-reads at the poly(A) tail resulted in many errors and poor consensus accuracy and highlights the challenges of analysing low complexity sequences with short-read sequencing (Fig. 3c).

## Analysis of mRNA integrity

Integrity directly impacts the effectiveness of an mRNA vaccine or therapy[24]. mRNAs can be fragmented due to hydrolysis, degradation by RNases, or abortive transcription, and do not encode a full-length, open reading frame that can be translated into an effective drug. Due to its use of full-length nanopore sequencing, VAX-seq can measure the length of mRNA vaccines and provide a quantitative measurement of an mRNA sample's integrity (Fig. 2c). We measured the size profile of the eGFP mRNA, which showed a primary peak (77%) that was within 5% of the expected length (1153nt), as well as a diverse range of smaller, fragmented mRNAs (that collectively comprise 23% of this mRNA sample). This size distribution profile calculated from the read-length is analogous to measurement using electrophoretic methods (Agilent TapeStation; Fig. S5c). However, an advantage of RNA sequencing is that individual peaks can be analysed to determine the mRNA sequences. We also analysed three replicate libraries, demonstrating the reproducibility of the VAX-seq workflow to measure mRNA integrity (Fig. S5d).

cDNA library preparation during the VAX-seq protocol adds two flanking adaptors to each 5' and 3' end of the mRNA. By analysing sequenced reads that include both flanking adaptors, we can distinguish full-length mRNA molecules from truncated mRNA molecules. For the eGFP mRNA, we found 58.2% of sequenced reads encompassed the full mRNA length (including the Kozak sequence, coding sequence and 3' UTR). Of the fragmented reads, 7% of reads were 3' truncated likely due to abortive transcription. The majority of the remaining sequences (31.3%) were 5' truncated likely due to degradation by RNases. The remaining 3.5% of reads displayed both 3' and 5' truncation and likely resulted from RNA hydrolysis. Determining whether the truncated fragments formed during in vitro transcription or library preparation will require the future development of vaccine sequencing standards. Short-read sequencing showed poor and uneven coverage that precluded an analysis of mRNA length and integrity (Fig. 3a). This heterogenous alignment coverage was highly reproducible between replicates ($R^2 = 0.99$, Fig. S8a, b). This demonstrates how long-read sequencing can provide a quantitative profile of the integrity of any mRNA vaccine sample during manufacture, storage or delivery.

## Characterising off-target RNA contaminants

In vitro transcription can generate off-target RNAs, including truncated, readthrough or antisense RNAs that can elicit an innate immune response and must be removed to ensure the safety and efficacy of mRNA vaccines[5–7]. We used VAX-seq to characterise these fragmented and off-target RNA contaminants in the cDNA libraries (Fig. 4e). To permit the detection of non-polyadenylated RNA, template DNA and sheared fragments, we performed an additional polyadenylation step of all RNA 3'-termini prior to library preparation (see Methods), which increased the number of off-target reads detected (Fig. S5e). The majority of sequences (92.7%) aligned to the on-target mRNA product, with few reads (0.01%) detecting *E. coli* contamination (Fig. S7a, c). The remaining (7.3%) of RNA species comprised different off-target RNAs. Of these, 0.3% were likely derived from cryptic transcription start sites (Fig. S11a–c). Further putative cryptic transcription start sites were detected in libraries with an additional de novo polyadenylation step, where transcription terminates upstream of the poly(A) tail (Fig. S12). These reads may have been derived from residual plasmid DNA template, however, as they align non-randomly across the plasmid reference sequence, they are likely the result of spurious in vitro transcription. The impact of cryptic promoters, and their contribution

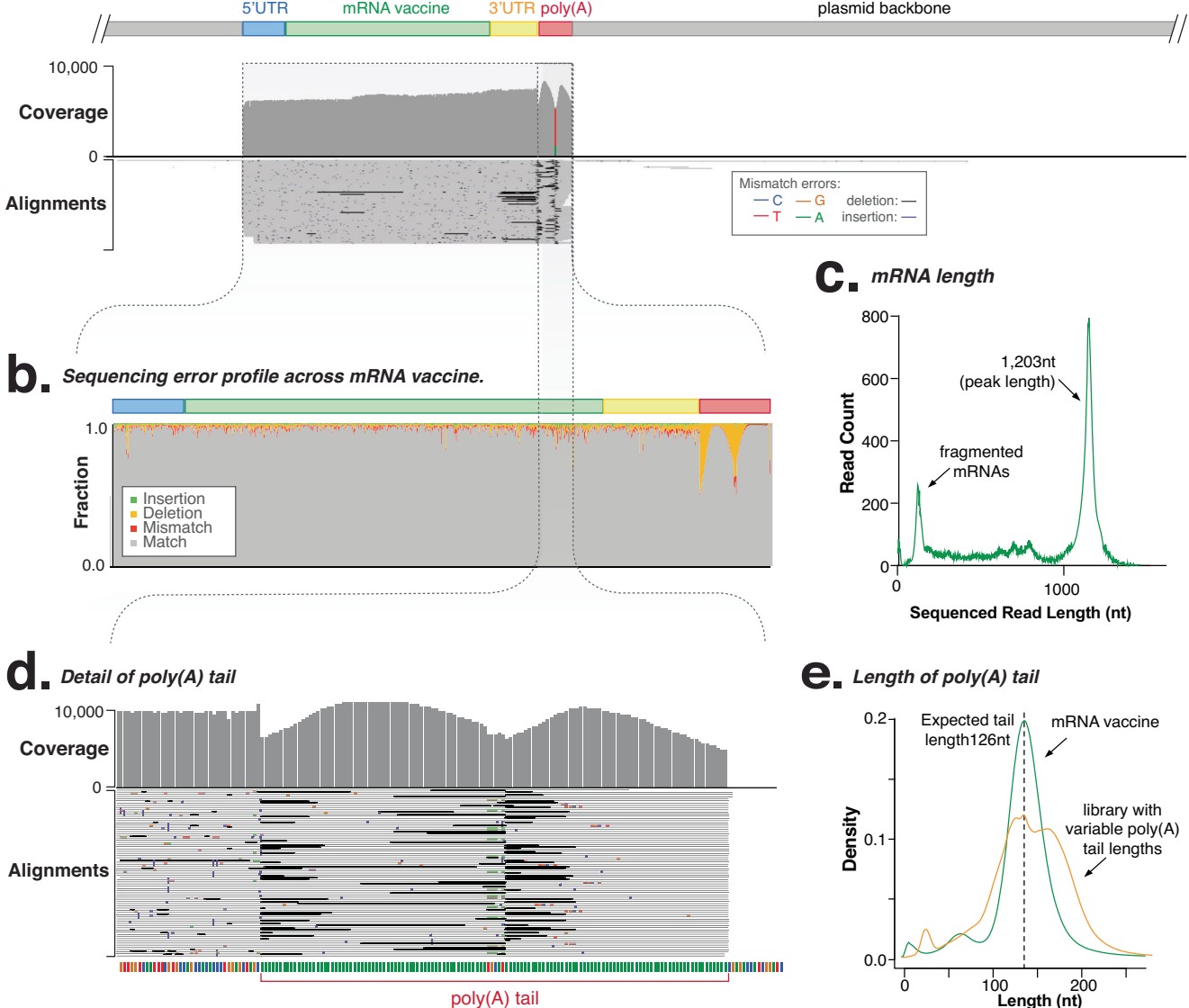

**Fig. 2 | Analysis of reference eGFP mRNA vaccine using long-read Oxford Nanopore sequencing (PCS111). a** Genome-browser (IGV) view of long-read cDNA alignments to the reference plasmid sequence. Coverage indicates the number of reads at each nucleotide position while the lower alignments grey bars indicate unique, individual alignments, with colouring indicating their similarity to the reference genome. **b** Sequencing error and type profile across mRNA vaccine and poly(A) tail sequences. **c** Detail shows sequencing coverage and error at the poly(A) tail, showing a characteristic m-shaped deletion profile. **d** mRNA length as measured using ONT full-length sequencing shows full-length and fragmented mRNA vaccines. **e** poly(A) tail length measured using *tailfindr* for eGFP mRNA (green), compared to a cDNA library with variable poly(A) tail lengths. Source data are provided as a Source data file.

to off-target RNA species can be further investigated by techniques such as DNA footprinting.

For comparison, we also analysed synthetic mRNAs with short-read (Illumina) cDNA sequencing. This method uses random hexamer priming and can sensitively detect non-polyadenylated RNA, such as truncated or antisense RNAs (Figs. 4e and S7b), which were likely derived from aberrant transcription, initiated 3' to the poly(A) tail[7]. Short-read sequencing also detected similar off-target RNAs, including upstream (1.4%) and downstream (5.7%) sequences (Fig. S7c, d). Analysis of short-read stranded orientation also allows the sensitive detection of ~0.6% of antisense RNA transcripts that can form extended stable dsRNA (Fig. S7d). This method for the indirect detection of dsRNAs may provide an alternative to dsRNA immunoblotting, the current industry standard technique for dsRNA detection (Fig. S7e)[7].

**Direct RNA sequencing of mRNA vaccines**

Nanopore sequencing enables direct sequencing of synthetic mRNAs without the need for reverse transcription and amplification steps during library preparation, and can directly analyse nucleoside modifications[25]. We performed direct RNA sequencing (SQK-RNA002), followed by analysis (see 'Methods', Fig. 4a). The observed yield of direct RNA sequencing libraries was lower (less than ~15% of sequencing yield per pore) than for matched cDNA sequencing libraries, and they cannot currently be multiplexed, which should be considered if performing large-scale sequencing during mRNA manufacture (Fig. S9a).

We first analysed mRNA vaccine quality using direct RNA sequencing, excluding the poly(A) tail. While providing a consensus sequence of sufficient quality, direct RNA sequencing had

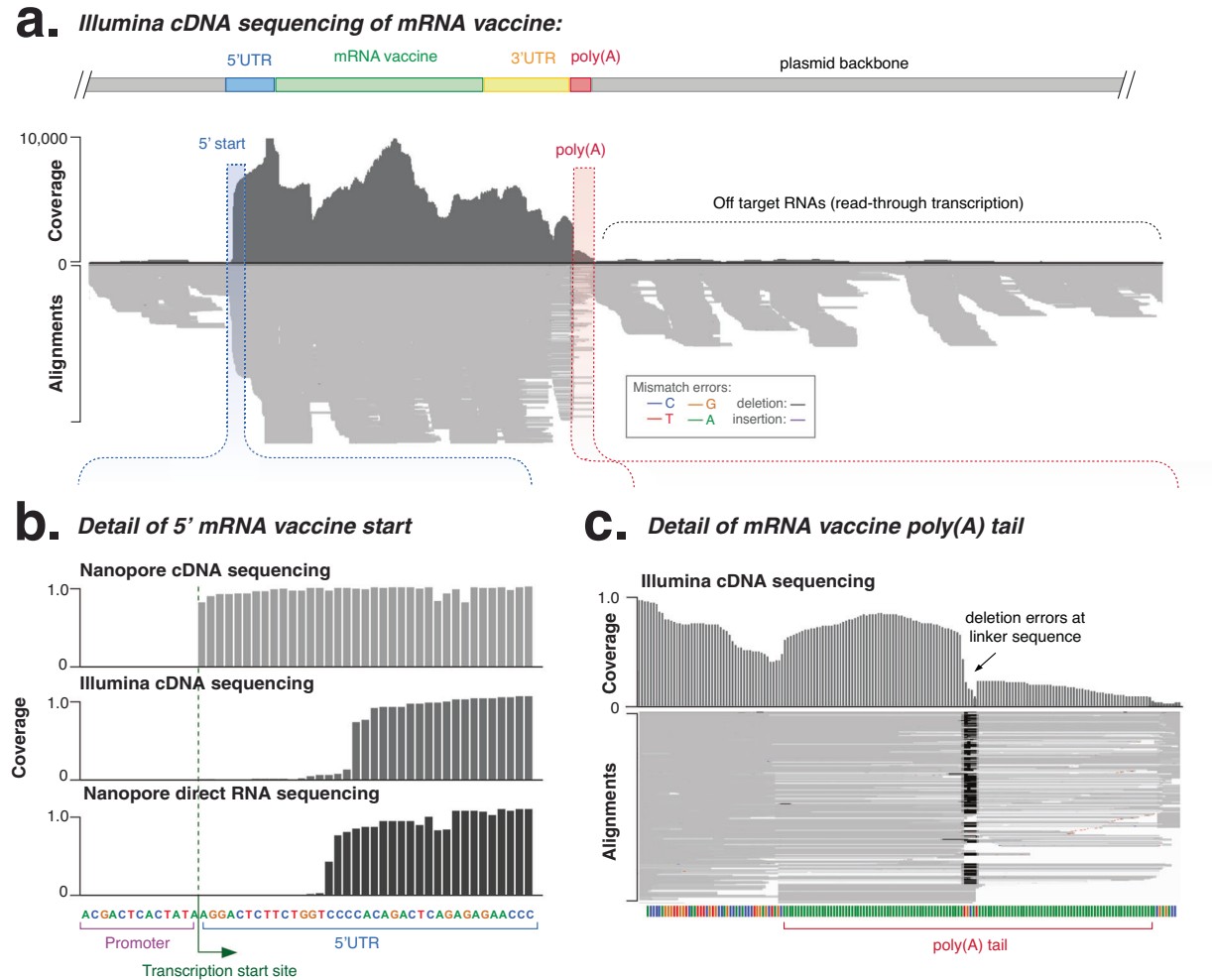

**Fig. 3 | Analysis of reference eGFP mRNA vaccine using short-read Illumina sequencing (TruSeq). a** Genome-browser (IGV) view of short-read cDNA sequenced libraries aligned to the reference plasmid sequence. Coverage indicates the alignment depth at each nucleotide position, while the lower alignments grey bars indicate unique, individual alignments, with colouring indicating their similarity to the reference genome. **b** Detail of transcription start site shows alignment of long- and short-read sequencing at 5' end of mRNA vaccine. **c** Detail genome browser view of poly(A) tail shows uneven coverage and deletion of linker sequence due to short-read misalignment. Source data are provided as a Source data file.

a higher mean error rate (7.78%, resulting from 2.61% mismatch, 5.18% deletion and 1.43% insertions) than matched cDNA sequencing (Figs. 4b and S9c). However, by comparing the per-nucleotide error rate against matched plasmid and cDNA sequencing, we could distinguish biases specific to direct RNA sequencing, including the deletion of low-quality poly(A) tail nucleotides (Fig. S9b). These deletion errors in the poly(A) tail likely caused a secondary peak in our mRNA length analysis, however, the analysis of the poly(A) tail length using *tailfindr* (which inputs raw nanopore sequencing data) corrected for this artefact and estimated a mean poly(A) tail length of 125.03nt (compared to the expected 126nt length; Fig. 4c, d).

**Detection of modified nucleosides in mRNA vaccines with direct RNA sequencing**
The incorporation of modified nucleosides into mRNA vaccines can reduce the innate immune response and improve the translation and stability of mRNA vaccines[26,27]. We performed RNA sequencing of mRNA vaccines that include N1-methylpseudouridine (see 'Methods'). We prepared both short- and long-read sequencing libraries from the modified mRNA. These libraries had lower yields (~50%) than matched, native mRNA vaccines, suggesting the modified nucleosides reduced the efficiency of cDNA library preparation (Fig. S9d).

We next analysed the impact of modified nucleosides on mRNA quality attributes. We found little impact on cDNA sequencing error between mRNAs incorporating native uridine and N1-methylpseudouridine, while direct RNA sequencing showed a higher error rate (Fig. 5a, b). Both cDNA and direct RNA sequencing showed that modified mRNA vaccines included more truncated transcripts, with fewer full-length (41%), and more (54%) truncated mRNA molecules, particularly below 500nt in length (Fig. 5c). Transcript size distribution was also compared using capillary electrophoresis, for vaccines incorporating N1-methylpseudouridine and unmodified bases, with minor fragment size differences observed between the modified and unmodified vaccines (Fig. S10a, b).

Direct RNA sequencing can detect modified nucleosides in single mRNA vaccine molecules[16]. A per-nucleoside comparison showed that direct RNA sequencing identified N1-methylpseudouridine nucleosides with a characteristic base-calling error that misclassified N1-methylpseudouridine as cytosines (0.62C/0.38U, Fig. 5d, e). These were low-confidence measurements and the error rate increased to 23.6% (11.8% with mismatch, 9% deletion, 2.8% insertion) at modified nucleosides, with this enrichment in deletion errors resulting in a shifted mRNA length profile (Fig. 5c). The consistent error profile at N1-methylpseudouridine suggests that retraining the base-caller will permit accurate detection of modified nucleosides[16].

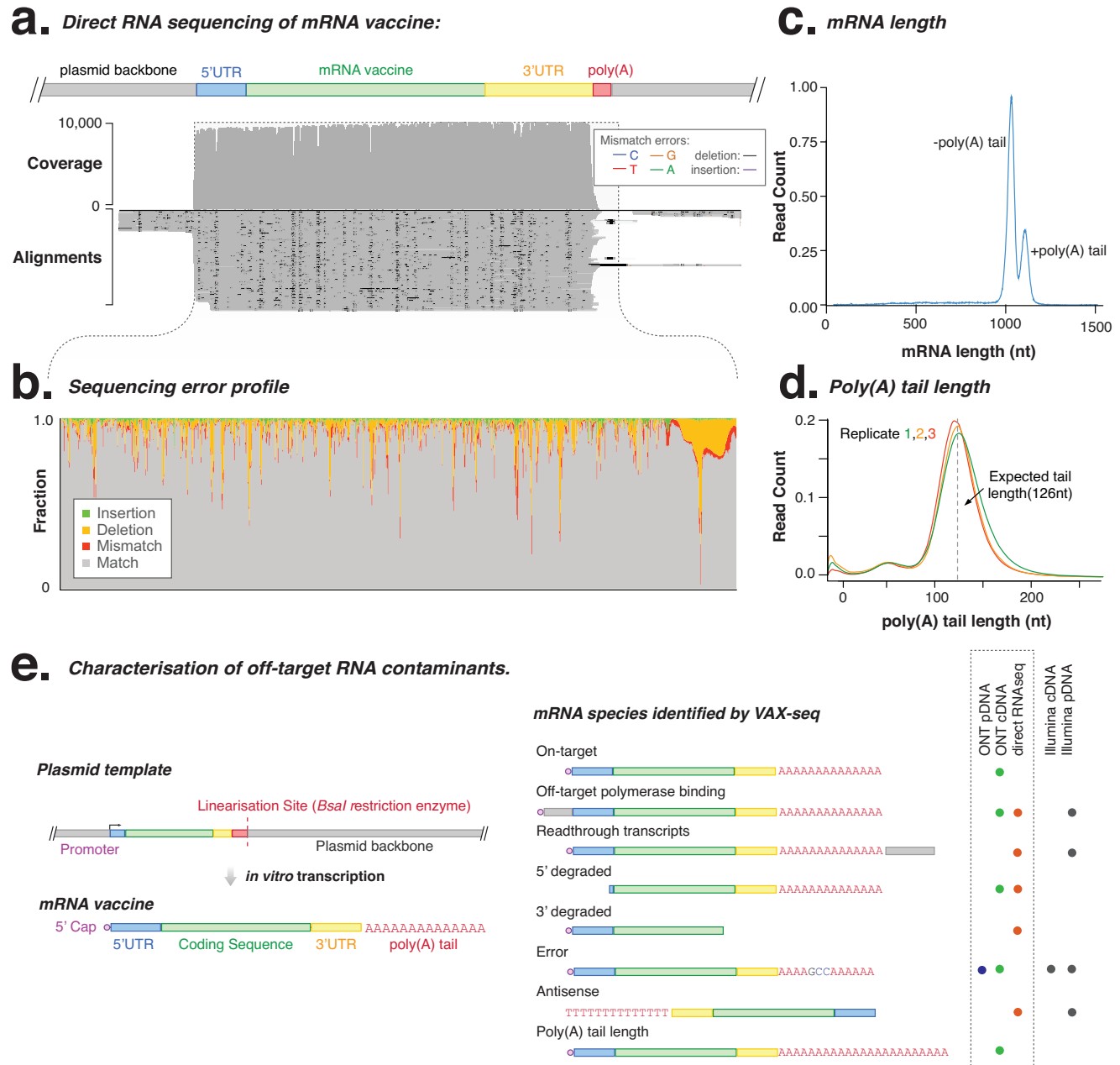

**Fig. 4 | Direct RNA sequencing of reference eGFP mRNA vaccine. a** Genome-browser (IGV) view of direct RNA sequencing alignments to the reference sequence. Coverage indicates the number of reads at each nucleoside position while the lower alignments grey bars indicate unique, individual alignments, with colouring indicating their similarity to the reference genome. **b** Direct RNA sequencing shows error type and frequencies across the mRNA vaccine and poly(A) tail sequence. **c** Plot shows the length of mRNA vaccine as measured from direct RNA sequencing, showing additional smaller peak resulting from artefactual trimming of poly(A) tail from reads. **d** Poly(A) tail length measured using *tailfindr* across three technical replicate direct RNA sequencing libraries. **e** Schematic diagram shows the different mRNA species, fragment size and contaminations identified by the VAX-seq workflow. Source data are provided as a Source data file.

## Discussion

Advances in manufacturing have enabled billions of mRNA vaccine doses to be produced with sufficient purity, quality and safety during the COVID-19 pandemic[9,28]. However, the analytical methods needed to measure the quality of mRNA vaccines are evolving. Here, we describe VAX-seq, a nanopore long-read sequencing protocol able to measure key mRNA quality attributes, including sequence identity, integrity and contamination. VAX-seq can measure mRNA quality at different manufacturing steps, from initial plasmid preparation to final product characterisation, providing a single comprehensive, and integrated analysis.

Within our study, we evaluated a range of different long- and short-read RNA sequencing methods to determine the best-practise workflow for analysing mRNA vaccines and therapies. Full-length cDNA sequencing using Oxford Nanopore chemistry provided several advantages. Nanopore cDNA sequencing was sufficient to confirm sequence identity with full and homogenous coverage of the entire mRNA vaccine length. This enables analysis of full-length and fragmented mRNAs resulting from degradation, which is a key mRNA attribute that directly limits effectiveness[24]. The VAX-seq protocol also uses a 3' reverse adaptor that ligates to the poly(A) tail end, allowing accurate measurement of the full poly(A) tail length. Additional

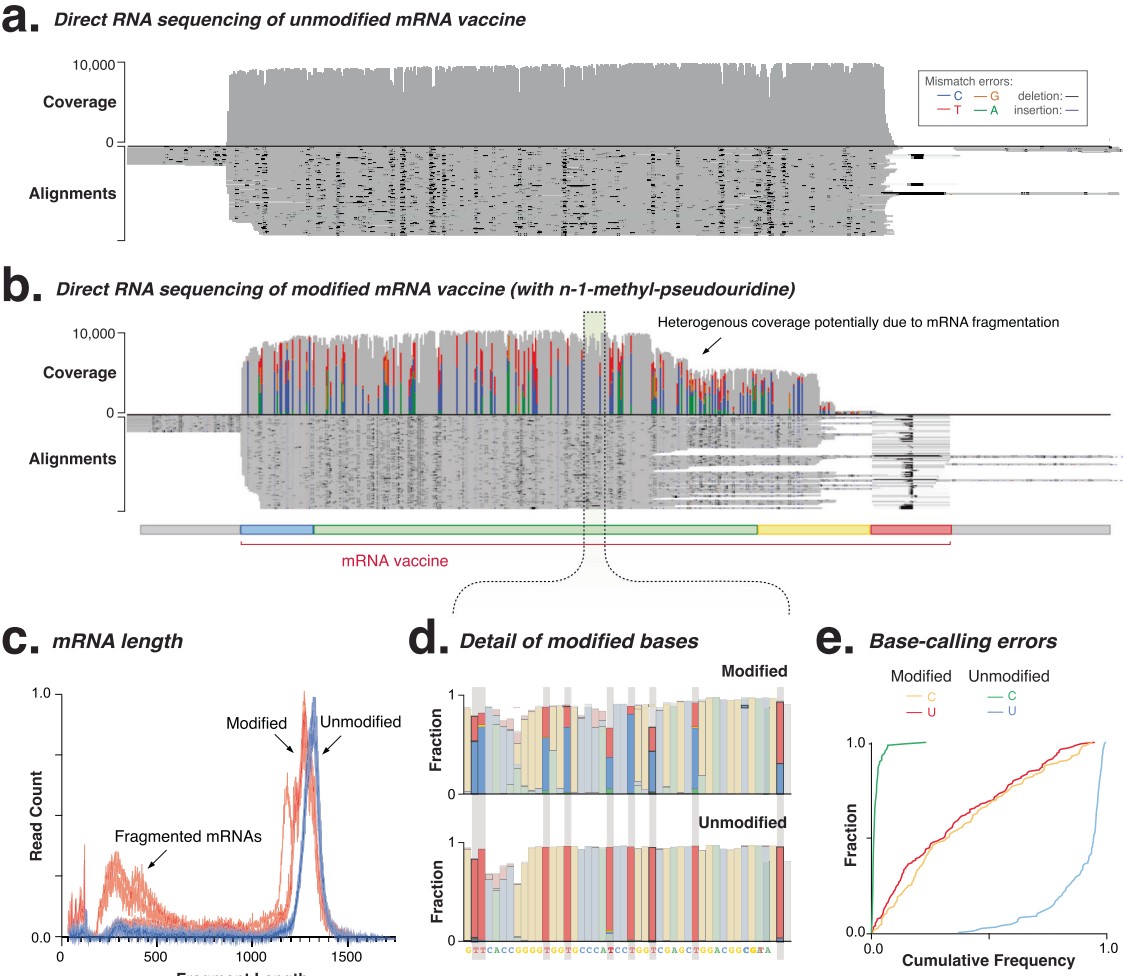

**Fig. 5 | Analysis of reference eGFP mRNA vaccine incorporating modified nucleosides using direct RNA sequencing. a, b** Genome-browser (IGV) view of long-read (ONT) alignment to the plasmid reference, sequenced using direct RNA sequencing for mRNA vaccines prepared with **a** Uridine and **b** N1 methylpseudouridine. Coverage indicates the number of reads at each nucleoside position while the lower alignments grey bars indicate unique, individual alignments, with colouring indicating their similarity to the reference genome. Heterogenous coverage is observed in the direct RNA sequencing, likely due to fragmentation of the modified mRNA vaccine. **c** mRNA length analyses demonstrate shorter length for modified mRNA vaccines due to fragmentation and enrichment in deletion sequencing errors ($n = 4$). **d** Direct RNA sequencing shows stereotypical errors (cytosine, blue; uridine; red) at N1-methylpseudouridine compared to uridine nucleosides. **e** Cumulative distribution plot shows per-nucleoside error profile for N1-methylpseudouridine (red/orange) compared to unmodified nucleosides (blue/green). Source data are provided as a Source data file.

poly(A) tailing of mRNAs reveals diverse off-target reads, which can be derived from the plasmid template. While we found cDNA sequencing more reliable than direct RNA sequencing, we nevertheless anticipate direct RNA sequencing will be a useful research tool to analyse mRNA vaccine chemistry, including the incorporation of modified nucleosides.

Within our study, we compared VAX-seq to current industry-standard techniques, including chromatography, capillary and agarose electrophoresis and immunoblotting. VAX-seq was shown to provide a quantitative and sensitive measure of mRNA features, providing a streamlined alternative to the currently recommended analytical techniques. For example, current industry standard techniques include immunoblotting to detect dsRNA, and capillary electrophoresis and RP-HPLC for mRNA integrity analysis. VAX-seq provides both antisense RNA detection (an indirect indicator of dsRNA) and mRNA integrity analysis, alongside sequence confirmation, in a single step protocol that can be performed in real-time. This enables rapid testing within hours of manufacture so quality control issues can be quickly detected to inform trouble-shooting.

Integrity and purity are two key features of mRNA vaccines that directly impact their effectiveness and adverse immune responses.

VAX-seq can routinely detect and characterise the complex off-target RNA contaminants that are produced during in vitro transcription. Similarly, long-read VAX-seq can also detect the shearing or degradation of mRNA vaccines that can occur during manufacture, storage and transport. Fortunately, VAX-seq requires little mRNA input and can be multiplexed to enable the affordable large-scale validation of many vaccine batches.

The recent success of the COVID-19 mRNA vaccines has focussed considerable attention and investment on the development of further mRNA vaccines and therapies[29]. However, to realise this potential, mRNA vaccines must be manufactured at the required quality and purity to ensure their safety and effectiveness[19]. Here we demonstrated the advantages of RNA sequencing methods, such as VAX-seq, for supporting the manufacture, quality control and development of new mRNA vaccines and therapies.

## Methods
### Design and synthesis of reference plasmid
A reference construct was first designed, with the intention of optimising the production of RNA therapeutics for pre-clinical research. The coding sequence of eGFP[30] was selected as a reporter in the coding

region, as its protein product can be assayed simply through Flow cytometry and other fluorometric methods. Transcription is initiated using a modified T7 promoter (5′-TAATACGACTCACTATAAGG-3′), which is compatible with CleanCap AG reagents (TriLink BioTechnologies) for co-transcriptional capping during mRNA IVT, and the 5′ and 3′ UTR sequences were selected from Human alpha globin and AES-mtRNR1 respectively, due to their demonstrated effect on mRNA stability and expression[31–33]. A 126nt segmented poly(A) tail was selected, as it was predicted to reduce the occurrence of plasmid recombination in *E. coli* during plasmid propagation and cloning, without compromising mRNA half-life and efficiency of translation in mammalian cells[17]. Our construct was synthesised and cloned into a pUC57-Kan backbone (GenScript), which contains an origin of replication and a kanamycin resistance gene.

Plasmid DNA was transformed using heat shock into NEB Stable competent *E. coli* cells (NEB, C3040H). A single colony containing the plasmid was then amplified in LB media with 30 μg/mL Kanamycin. *E. coli* was harvested from 2500 mL of culture using centrifugation and lysed using alkaline lysis. The supercoiled plasmid DNA was then purified using multistep FPLC purification, first using anion exchange chromatography, and then using a desalting column, where the buffer was exchanged for TE. One mg of supercoiled plasmid DNA was linearised using BsaI-HFv2 restriction enzyme (NEB, R3733), generating a single fragment that terminates at the end of the poly(A) tail. This linearisation ensures run-off in vitro transcription, terminating immediately 3′ to the poly(A) tail. After linearisation, the plasmid DNA was purified further with hydrophobic interaction chromatography (HIC). The linearised plasmid DNA was diluted with 3 volumes of 4 M ammonium sulfate, then purified using HIC to remove other isoforms of plasmid DNA. The pooled fractions corresponding to linear plasmid DNA were buffer exchanged with TE Buffer on a desalting column to remove the ammonium sulfate. The purified linear plasmid DNA was concentrated using isopropanol precipitation to an approximate concentration of 500 ng/μL.

We evaluated the length and purity of our linearised plasmid template using a range of different analytical methods. First, the size distribution of the linearised plasmid was compared to supercoiled plasmid using Agarose gel electrophoresis. This permits the analysis of the proportion of linearised plasmids in the preparation. Second, the purity of the supercoiled and linearised plasmid DNA was analysed by HPLC using a CIMac™. The pDNA was analysed on a 0.3 mL (1.4 mm) diethyl-aminoethyl (DEAE) weak anion-exchange column (BIA Separations, Adjovščina, Slovenia) connected to a PATfix® analytical HPLC system (Sartorius, Goettingen, Germany). HPLC was performed using mobile phase Buffer A (0.1 M Tris, 0.3 M Guanidine-HCl, 1% Tween-20 (w/v), pH 8.0) and Buffer B (0.1 M Tris, 0.3 M Guanidine-HCl, 0.7 M NaCl, 1% Tween-20 (w/v), pH 8.0).

## Oxford Nanopore plasmid sequencing

Oxford Nanopore sequencing was used to measure the accuracy and purity of plasmid template preparations. Ligation Sequencing libraries (SQK-LSK109) were prepared from linearised plasmid templates (described above) and were labelled with PCR-Free Native Barcodes (EXP-NBD104). Libraries were prepared according to manufacturer's instructions (Oxford Nanopore Technologies), with the exception that 2 μg of template was used as an input rather than the recommended 1 μg. The resulting libraries were quantified on a Qubit instrument (Invitrogen) with the dsDNA HS kit and qualitative analysis of fragment length distribution was completed with D5000 Screen-Tapes (Agilent Technologies, USA). The results of the quantitative and qualitative analyses were used to calculate the required library concentrations for pooling and loading. Barcoded libraries were sequenced on either R9.4.1 (FLO-MIN106D) or Flongle Flow Cells, with High Accuracy live base-calling enabled (Guppy v5.1.13 and MinKNOW

Core 4.5.4). All nanopore reads with quality scores >9 were used to form a concatenated FASTQ file and proceeded to further analysis.

## Bioinformatic analysis of Oxford Nanopore plasmid sequencing

First, barcodes were removed with Porechop v0.2.4 Oxford Nanopore pDNA sequencing data were then mapped and analysed using a custom pipeline. The quality-filtered, concatenated FASTQ reads were aligned to the plasmid reference via Minimap2 (Release 2.20-r1064) with -ax map-ont for Nanopore[34]. The resulting SAM alignment files were processed via SAMtools v1.15 to generate sorted and indexed BAM files, as well as various other mapping analysis files[35]. The generated BAM files were viewed and analysed utilising Integrative Genomics Viewer (IGV v2.12.3)[36]. Further run and sample quality statistics were acquired via NanoPlot v1.38.1 and pycoQC v v2.5.2[37,38].

The per-nucleotide error profile of mapped reads was determined relative to the reference index sequence using pysamstats v 1.1.2 with options --max-depth=300000000 --FASTA --type variation (https://github.com/alimanfoo/pysamstats). To perform simple error correction, the per-nucleotide error profile of the plasmid sequences was subtracted from the corresponding nucleotides within the cDNA/dRNA sequences. Plotting and statistical analyses were performed using Excel (v 16.67 for Mac) and GraphPad Prism (v 9.3.1 for Mac) software.

To investigate the content of the unmapped reads for possible contamination a BAM file was generated for unmapped reads via SAMtools (v1.15) utilising SAMtools view -S -b -f 4 and this was converted to a FASTA file via SAMtools FASTA. This FASTA file was aligned to an *E.coli* reference sequence utilising Minimap2 then sorted and indexed BAM files and alignment statistics were generated with SAMtools as described earlier. Sequence homology of the reads that did not align to the *E.coli* reference was investigated using the Basic Local Alignment Search Tool (BLAST) Nucleotide Collection nr/nt provided by the National Center for Biotechnology Information.

Variant calling and consensus sequence generation was completed using bcftools (v1.15)[35] with the following set of commands comparing the BAM file to the reference and generating a VCF; bcftools mpileup -d 300000000 --no-BAQ --min-BQ 0 -Ou -f | bcftools call -c -M --ploidy 1 -Oz -o *.vcf.gz. The resulting VCF was indexed and normalised to the reference and a BCF generated. A consensus.fa sequence was generated after comparing the VCF to the reference sequence with command bcftools consensus and option -a – to replace positions absent from the VCF (zero coverage) with a character.

## Illumina plasmid DNA sequencing

Plasmid DNA templates were next sequenced on an Illumina MiSeq instrument. Barcoded Illumina DNA PCR-free libraries were prepared from the same linearised plasmid template used for ONT sequencing, according to the manufacturer's instructions (Illumina). The resulting libraries were sequenced on a MiSeq instrument at the Australian Genome Research Facility using v2 chemistry, set to 150 base paired end.

BCL files generated on the Illumina MiSeq were processed with Illumina DRAGEN BCL Convert 07.021.609.3.9.3 pipeline to generate FASTQ.gz files. The quality of the reads within these files were checked with FastQC (v0.11.9). Reads were then adaptor and quality trimmed utilising TrimGalore (v0.6.8dev) to create FASTQ.gz files with reads passing the Q20 threshold[39]. Adaptors were trimmed from the Illumina PCR-Free libraries using --stranded_illumina --paired. Additionally, T overhangs were removed from the Illumina stranded mRNA libraries using --clip_r1 1 --clip_r2 1. The resulting trimmed files were checked with FastQC and then aligned to an indexed plasmid reference file[40] using BWA-MEM (bwa-0.7.17-r1188).

## In vitro transcription of mRNA

Capped mRNA with modified nucleotides was produced by in vitro transcription (IVT) using T7 RNA polymerase following protocols described in Henderson et al.[41] and according to the manufacturer's instructions (NEB, E2080S). Briefly, 50 µg/mL purified linear plasmid DNA was used as template for an IVT reaction at 32 °C for 3 h with 16 µg/mL T7 RNA polymerase (NEB M0251), ribonucleotides (6 mM ATP, 5 mM CTP, 5 mM GTP; NEB, N0450), 5 mM N1-methylpseudouridine-5′-triphosphate (TriLink BioTechnologies, TRN108110), or 5 mM UTP for matched unmodified controls, transcription buffer (40 mM Tris·HCl pH 8.0, 16.5 mM magnesium acetate, 10 mM dithiothreitol (DTT), 20 mM spermidine, 0.002% (v/v) Triton X-100), 2 U/mL yeast Inorganic pyrophosphatase (NEB, M2403) and 1000 U/mL murine RNase inhibitor (NEB, M0314). Cap1 analogue was co-transcriptionally incorporated to the mRNA 5′ end by addition of 4 mM CleanCap AG reagents (TriLink, TRN711310) to the reaction. The mRNA IVT reaction was stopped by addition of 200 units DNaseI (NEB, M0303) per mL of IVT reaction and incubation at 37 °C for 15 min. The resulting mRNAs were purified using Monarch RNA cleanup kits (NEB, T2050) according to the manufacturer's instructions with the final elution in distilled ultrapure water (ThermoFisher Scientific, 10977015).

We evaluated the yield, length and purity of the IVT mRNAs using a range of different analytical methods. mRNA was quantified by UV spectrophotometry analysis using a NanoPhotometer N120 (Implen) and the size distribution was evaluated using TapeStation electrophoresis with RNA ScreenTapes (Agilent Technologies, USA, 5067-5576).

## Detection of dsRNA contaminants by dot blot analysis

The IVT mRNA samples were diluted in nuclease-free water to final concentrations of 200, 500, 1000 and 2000 ng/µL. From the diluted samples, 5 µL aliquots were loaded onto a positively charged nylon membrane (Roche, Basel, Switzerland) which resulted in a total loading amount of 1000, 2500, 5000 and 10,000 ng of IVT mRNA samples on each blot respectively. dsRNA was manufactured in-house[42] as a positive control and nuclease-free water was used for the negative control. Samples were loaded onto a Bio-Dot® Microfiltration Apparatus (Bio-Rad, CA, USA) as per manufacturer's instructions. The membrane was air dried, then blocked by immersion and incubated with blocking buffer containing 5% non-fat dry milk in TBS-T (50 mM Tris-HCl, 150 mM NaCl, 0.05% Tween (w/v)) at room temperature for 1 hour with agitation.

For the detection of dsRNA, the membrane was incubated overnight at 4 °C with two different dsRNA-specific murine monoclonal antibodies (mAb), which were derived from clones 3G1 and 2G4 (Mozzy Mabs, Brisbane, Australia). Both antibodies were incubated separately overnight at a 1:5 dilution, diluted in incubation buffer (1% Non-Fat Dry Milk in TBS-T). The membranes were then rinsed 3 times then washed 3 times for 15 min each wash with TBS-T. The membranes were then incubated with horseradish peroxidase (HRP)-conjugated goat anti-mouse immunoglobulin G (IgG) secondary antibody (Abclonal, MA, USA) at 1:5000 dilution, diluted in incubation buffer, for 1 h with agitation. The membranes were then rinsed 3 times then washed 3 times for 15 min each wash. Chemiluminescence detection was performed using Novex™ ECL chemiluminescent substrate kit (Invitrogen, MA, USA) and the signal intensities of the dots were visualised using the ChemiDoc MP Imaging System (Bio-Rad, CA, USA).

## ONT cDNA-PCR sequencing

cDNA-PCR sequencing was used to determine the accuracy and purity of the IVT mRNAs. First, mRNA concentration was calculated using the Qubit RNA BR kit (ThermoFisher Scientific). mRNAs were diluted in nuclease free water to an appropriate concentration for library preparation (~1 ng/µL), and concentrations were confirmed using a Qubit RNA HS kit (ThermoFisher). Barcoded ONT cDNA-PCR libraries

(SQK-PCB109 and SQK-PCS111) were prepared according to the manufacturer's instructions (Oxford Nanopore Technologies), with the following exceptions. Evaporation during the cDNA synthesis step was assessed by measuring reaction volume, and tubes were topped up with nuclease-free water where appropriate. The cDNA was amplified for 14–16 cycles (recommendation is 14–18 cycles). Finally, libraries were eluted in 8 µL of Elution Buffer (rather than the recommended 12 µL volume) to boost the final concentrations of the libraries. This was necessary, as libraries prepared with templates containing modified bases appear to produce lower output libraries than those prepared with unmodified bases.

The resulting libraries were quantified via a Qubit instrument (Invitrogen) with the dsDNA HS kit (ThermoFisher Scientific) and qualitative analysis of fragment length distribution was conducted using D5000 ScreenTapes (Agilent Technologies, USA). The results of the quantitative and qualitative analysis were used to adjust library concentrations for pooling and loading. Up to 10 barcoded libraries were sequenced on each R9.4.1 (FLO-MIN106D) Flow Cell, with High Accuracy live base-calling enabled (Guppy v5.1.13 and MinKNOW Core 4.5.4). All nanopore reads with a quality score >9 were allocated as passed and proceeded to further analysis.

## Bioinformatic analysis of ONT cDNA-PCR sequencing

cDNA-PCR data were analysed as described for ONT pDNA sequencing (see above). Additionally, the following analyses were conducted. To identify full-length FASTQ transcripts containing SSP and VNP primers in the correct orientation, the tool pychopper (v.2.5.0) was utilised (with default parameters). These adaptors were then trimmed and new FASTQ files were generated. Trimmed reads from rescued and full-length folders were merged and these reads were mapped as described above with Minimap2 to the plasmid reference. BAM files were also generated as described above and these were used for analysis of cDNA read lengths. SAMtools (v1.15) utilising SAMtools view -F 2048 was used to extract read length distributions of the primary aligned reads from the BAM files. Plotting and statistical analyses were performed using GraphPad Prism (v9.4.1 for Mac) software to create read length distribution profiles for each sample.

Next, we analysed the proportion of on-target reads from our cDNA-PCR dataset (i.e. reads that were correctly transcribed and are likely to be translated into a functional protein). First, BEDtools (v2.27.1) intersect was utilised to analyse the sorted BAM files and identify the proportions of on- and off-target reads. BED files were generated, indicating target coordinates that include the start of the Kozak sequence and end of the 3′ UTR. Reads were binned as on-target and a BAM file generated if they overlapped the start and stop coordinates all other reads were binned as off-target. Off-target reads were further filtered down depending on start or stop coordinate overlap indicating 3′ or 5′ degradation. The generated BAM files were viewed and analysed utilising Integrative Genomics Viewer (IGV v2.12.3).

Variant calling and consensus sequence generation of the mRNA encoding region was completed as described earlier except that it also utilises the option --targets to restrict the pileup to the start and stop coordinates specified above.

## poly(A) tail length calculation using ONT cDNA-PCR sequencing

Poly(A) tail length was estimated from cDNA-PCR data (SQK PCS111) which anchors the oligo dT primer to the 3′ end, and using tailfindr (v1.3) using protocols described in Kraus et al.[21]. Briefly, unaligned FAST5 files are inputted, which are then segmented into the sequencing adaptor, splint adaptor, poly(A) tail and gene body. poly(A) tail length is then calculated based on estimates of pore translocation time. Poly(A) tail length was estimated from Fast5 files generated from the original cDNA-PCR kit (SQK PCB109) and the updated version (SQK PCS111). Here, Fast5 files from both kits were basecalled with High Accuracy utilising the following configuration

dna_r9.4.1_450bps_hac.cfg (Guppy v5.1.13 and MinKNOW Core 4.5.4). Then, the find_tails function of *tailfindr* (v1.3) was run using default settings[21]. Fast5 files generated from the SQK PCB109 kit required specification of custom cDNA 5′ (TTTCTGTTGGTGCTGATATTGCT) and 3′ (ACTTGCCTGTCGCTCTATCTTC) primer details to enable estimation.

### Illumina cDNA sequencing
Barcoded Illumina stranded mRNA libraries were prepared from the same IVT mRNA templates used for ONT sequencing, according to manufacturer's instructions (Illumina). The resulting libraries were multiplexed with the plasmid DNA libraries and sequenced on a MiSeq instrument at the Australian Genome Research Facility using v2 chemistry, set to 150 base paired end. Trimming and mapping of our Illumina cDNA sequencing data followed the pipeline described for Illumina pDNA sequencing (see above).

The generation of mapping statistics, per-nucleotide error calculations, variant calling, consensus sequence generation and unmapped reads analyses were conducted as described above for ONT sequencing. For samples sequenced on the Illumina platform, on and off target reads were calculated via a different method due to the short length of the reads. SAMtools (v1.15) with the command SAMtools view (various coordinates) was utilised to extract reads down-stream and up-stream of the mRNA encoding region of the reference and generate corresponding BAM files. SAMtools flagstat was utilised to count the primary reads contained within these BAM files and this number was compared to the total primary reads to calculate on target read numbers.

To identify antisense reads we used SAMtools view to separate reads that originated from the forward (second in pair of forward strand, -b -f 128 -F 16 and first in pair of the reverse strand, -b -f 80) or reverse strand (second in pair if they map to the reverse strand, -b -f 144 and first in pair if they map to the forward strand, -b -f 64 -F 16). The reads originating from the reverse strand were designated as antisense reads.

### Modified polyadenylation protocol
To detect fragments that lack a poly(A) tail in our synthetic RNAs, we enzymatically added poly(A) tails to our eGFP mRNA. This was prepared according to an Oxford Nanopore protocol, using *E. coli* Poly(A) Polymerase. Briefly, up to 10 μg of RNA was incubated with NEB *E. coli* Poly(A) Polymerase (M0276) and 1 mM ATP for 1.5 min. The reaction was stopped using EDTA (to a final concentration of 10 mM), making a final volume of 25 μL. The poly(A) tailed RNA was then cleaned up using 72 μL of Agencourt RNAClean XP beads (Beckman Coulter A63987) and was eluted in 12 μL of nuclease free water. The eluate was then quantified using Qubit BR or HS RNA kits (depending on the quantity of RNA that was inputted into the reaction). An appropriate quantity of RNA was then used as a template for SQK-PCB109 or SQK-RNA002 library prep. Libraries were prepared and sequenced as described above.

### Direct RNA sequencing with Oxford Nanopore
Direct RNA sequencing was used to determine the accuracy and purity of the IVT mRNAs. Libraries were prepared using the ONT SQK-RNA002 kit according to the manufacturer's instructions (Oxford Nanopore Technologies), with the following exceptions. First, 400 ng of template was used to prepare each library, rather than the 500 or 50 ng recommended in ONT direct RNA-sequencing protocols. We utilised the cDNA synthesis step, which improves the template stability. Additionally, we used Superscript IV, rather than the recommended Superscript III, due to its higher efficiency. Libraries were quantified using a Qubit RNA BR kit (ThermoFisher Scientific). Each library was sequenced on a separate R9.4.1 (FLO-MIN106D) Flow Cell for 72 h. Fast5 files were later basecalled with High Accuracy utilising the following configuration rna_r9.4.1_70bps_hac.cfg (Guppy v5.1.13).

All nanopore reads with a quality score >9 were allocated as passed and proceeded to further analysis. Direct RNA sequencing data were analysed as for cDNA, with the exception of barcode detection using Pychopper, and on- and off target detection calculations. Moreover, poly(A) tail length calculations were conducted as described for SQK PCS111 libraries (above).

Next, poly(A) tail length was estimated from direct RNA sequencing data (SQK-RNA002). Poly(A) tail length was estimated using *tailfindr* (v1.3), using protocols described in Kraus et al.[21]. *tailfindr* (v1.3) utilises the Fast5 files to estimate tail length and this tool was initially run on the same direct RNA dataset as Nanopolish v0.13.3 poly(A). After comparison of the estimated poly(A) tail length, further samples were analysed. *tailfindr* was run with default settings, generating a.tsv file as output. This.tsv file was interrogated with a custom R script generating a density plot of the estimated poly(A) tail length for each read.

### Statistics and reproducibility
Data depicted in this paper are predominantly a proof of concept, and range from $n = 1$ to $n = 4$. Details on experimental replication is included in individual results sections. No statistical method was used to predetermine sample size. No data were excluded from the analyses. The experiments were not randomised, as template quality is the most significant determinant of sequencing quality, rather than any alternative co-variant. The investigators were not blinded to allocation during experiments and outcome assessment, as knowledge of sample identity was unlikely to bias the interpretation of the results.

### Reporting summary
Further information on research design is available in the Nature Portfolio Reporting Summary linked to this article.

## Data availability
The genomic and transcriptomic data generated in this study have been deposited in the Sequence Read Archive (SRA) under accession codes "PRJNA856796" for RNA-seq and "PRJNA854780" for plasmid re-sequencing. The *E. coli* genome used in this work is available from NCBI under accession number "NC_000913.3". Further data generated in this study are provided in the Source data file, with larger files available through DRYAD (https://doi.org/10.5061/dryad.s1rn8pkds). Source data are provided with this paper.

## Code availability
*Mana* software[43] is available at https://github.com/scchess/Mana, with no access restrictions, and a reference to the code version used in this study is available through https://doi.org/10.5281/zenodo.8190088.

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

## Acknowledgements

We acknowledge the following sources of funding and support: National Health and Medical Research Council (GNT2014002 and GNT1161832) to T.R.M., National Collaborative Research Infrastructure Strategy (NCRIS) to T.R.M. and T.M., Therapeutic Innovation Australia (TIA) to T.R.M. and T.M., Genome Innovation Hub (GIHEX22-VAC) to H.M.G. and The University of Queensland to T.R.M. and T.M. Illumina library preparation and sequencing was conducted by the Australian Genome Research Facility (AGRF), and Direct RNA sequencing libraries were prepared and sequenced and analysed at the Genome Innovation Hub. The contents of the published materials are solely the responsibility of the administering institution, a participating institution, or individual authors, and they do not reflect the views of the NHMRC, NCRIS or TIA.

## Author contributions

H.M.G., T.R.M. and S.I. conceived the study, with input from E.M., R.F., A.H. and T.M. The mRNAs used in this study were prepared by E.A., J.R.P. and R.T., and their quality was tested by D.J.H. using USP protocols. H.M.G. and S.K.R. prepared and sequenced the Oxford Nanopore libraries used in this study, and T.R.M., S.I., S.S., J.X. and S.W.C. analysed the data. S.I., T.R.M. and T.W. developed the code for the Mana software. The first draft of the manuscript was written by H.M.G., T.R.M. and S.I., with subsequent drafts reviewed by all authors.

## Competing interests

T.R.M., H.M.G. and S.W.C. have received financial support from Oxford Nanopore Technologies for travel and accommodations. The other authors declare no competing interests.
