## [Peer Review File · Nature Communications]

REVIEWER COMMENTS

Reviewer #1 (Remarks to the Author):

The manuscript by Gunter et al. seeks to address an important question in mRNA vaccinology: that is, how to study the complex mixture of RNA species that results from in vitro transcription of synthetic mRNA vaccines. This matters because double-stranded and other unintended RNA species can trigger innate immune sensing and inhibition of translation, not to mention issues for quality control and fragmented mRNA (during or after transcription). The unintended RNA species in mRNA vaccines are poorly characterized on the sequence level, and this area of research is likely to foster the development of superior (cheaper, faster, more reliable) methods of synthesizing mRNA vaccines. The authors use long-read nanopore sequencing as a promising technology to characterize these RNA species, comparing the data to short-read Illumina sequencing. As a disclaimer, my background is in mRNA vaccine design and immunology, and I do not have much background in the field of high-throughput sequencing. Therefore, I have focused my review on the value of this manuscript to the mRNA vaccine field.

The main strength that was demonstrated for the long-read sequencing method is the ability to detect truncations and other aberrant RNA species in a way that reveals the full length of each RNA species.

My enthusiasm for the value of this contribution was diminished by the fact that it reveals little specific information about the actual RNA sequences present in the mRNA vaccine samples, and a number of other scientific concerns detailed below.

Scientific concerns:

1.) Many of the figures show sequence coverage/alignment but lack specific information about the identity of the RNA species in question, in particular the unintended RNAs, which would be much more useful. Fig. 4e begins to touch on this, but the information presented here could be greatly expanded and made more quantitative and specific. In order to translate these results to the improvement of RNA vaccine design, we would need to see the actual aberrant RNA species that are present (sequence, length, etc). Where are the cryptic transcription start sites? Where is antisense transcription initiating from? What dsRNA species could potentially form? Are dsRNA species directly detectable using this method? All this will lead to models about how these aberrant RNA species are generated and how to avoid them if necessary. In addition, many figures contain elements (in particular black, gray, white, and green colors) that are not defined and render it inaccessible to readers, especially those who are not deep in the field of high-throughput sequencing.

2.) What are the details of the COVID-19 mRNA vaccines described here? Pfizer/BioNTech vs. Moderna? How were these obtained and treated? This information is necessary for these parts of the paper to be interpretable.

3.) Can we distinguish the source of errors in mRNA sequence? (T7 transcription errors vs plasmid-derived errors vs hydrolysis after transcription, etc.)

4.) I am concerned about the detection of a 66 nt deletion in the poly(A) tail of one of the COVID-19 vaccines using the tailfindr method, while no truncation was detectable by capillary or agarose electrophoresis (line 126). This appears to challenge the validity of their method.

5.) What explains the different rates of identification of upstream, downstream, and antisense sequences between long read and short read methods, or is this discrepancy simply stochastic?

6.) Fig. 5 shows a large fraction of truncated mRNAs identified by long read sequencing, yet this is not visible in the capillary electrophoresis shown in Fig. S11a, appearing to contradict what the authors state on line 206. If the nanopore sequencing does not agree with electrophoresis, does that mean there can be fragmentation artifacts in nanopore sequencing (hydrolysis, shearing, aborted sequencing, etc?), and might this be more likely with N1-methylpseudouridine?

7.) Given the error rate of 23.6% reported for N1-methylpseudouridine (line 213), it is unclear how robust the base-calling error method is for identifying this modified nucleoside. Also, if the error rate is higher for N1-methylpseudouridine compared to other nucleosides, it is unclear whether this is due to an inaccuracy of sequencing or perhaps an error rate in T7 transcription.

Other concerns:

1.) What is the positive control for dsRNA in the dot blot?

2.) What is a “pre-nucleotide comparison” (line 208)?

3.) It is unclear which authors have the competing interest from Oxford Nanopore Technologies as there are no middle initials given in the author list and in some cases the authors have the same first and last initials.

4.) Figure 2 – is this all for the eGFP mRNA? What is the negative polyadenylated control?

Reviewer #3 (Remarks to the Author):

The authors of “Analysis of mRNA vaccines and therapies using VAX-seq” used Nanopore sequencing to evaluate RNA samples at various steps during the manufacturing process to assess the Quality of the samples. They demonstrated that long reads were valuable for determining if the linearization process was completed, and if the IVT synthesis from those plasmids was successful. Gunter and Idrisoglu et al. found that Nanopore sequencing was useful for detecting IVT fragments and that both short-read and Nanopore sequencing were beneficial for detecting IVT synthesis contaminations, especially low abundance antisense products. The authors also evaluated short-read and Nanopore cDNA sequencing, and direct RNA nanopore sequencing on IVT synthesis products with N1-methylpseudouridine in place of canonical uridine, and found that although possible, direct RNA nanopore had much lower accuracy on these products.

This manuscript is well written, the experiments executed well, and the authors claims are supported by the evidence presented. Although it has not been previously demonstrated in the literature, this body of work represents an obvious but useful application of Nanopore sequencing. One thing that would help this manuscript is a description of how sensitive the sequencing needs to be to separate a good from a bad sample. For example, a sample that would pass QC by the orthogonal methods but fail when evaluated by sequencing would clearly demonstrate the advantage of using sequencing over the other established methods for mRNA vaccine QC.

Below are some specific points for the authors:

1.1 Adding citations about the safety of mRNA vaccines would improve the first paragraph of the introduction.

1.2 What did the other 2.2% of the reads align to if not the production strain genome when the antigenic sequences were used in the construct instead of the eGFP sequence. Also, do the authors know why there was substantially more contamination for the eGFP coding sequence than the antigenic sequences?

1.3 How did the authors detect circular plasmid sequences with nanopore sequencing? The plasmid must be linearized (such as by the restriction enzyme, sonication, or the rapid library kit) before it can be sequenced by nanopore and this was not clear.

1.4 Given that the IVT molecules are synthesized from 5' to 3' ends, I am surprised that such a high proportion of molecules (37%) were degraded from the 5' end. Was there a punctuated site where this degradation started/stopped, or was there a gradient? How many nucleotides were missing from the 5' end? If only 58.2% were full length, then why were 74% of the expected size? Does this mean that there was a contaminating nuclease that could bypass the m7G capped 5' end in the mRNA manufacturing process? Was this high rate of 5' end degradation also observed with the antigenic mRNA sequences? Can the authors elaborate further on this observation?

1.5 Figure 1, there is no (a) marker in the figure and referencing one in the legend seems unnecessary.

1.6 Figure 3, panel C is mislabeled as D in the figure legend.

1.7 Figure 4, a reference to panel A is missing.

1.8 When discussing the direct RNA nanopore sequencing error rate in the paragraph starting on line 183, was the error rate evaluated with and without including the poly(A) tail region in the reference? Much of the poly(A) tail should not have been basecalled by guppy and that may have increased the deletion error rate? It is not entirely clear from this paragraph how that was handled.

1.9 I couldn't find either of the two SRA accession IDs in the data accessibility section. To replicate the authors results, it is important to ensure that both the fastq and the fast5 files from the nanopore datasets are available for download because TailfindR was used.

1.10 Why was a quality cut off 9 used instead of 7, which is more typical of Nanopore experiments? Is it possible that more off-target reads can be found in the lower quality reads? Was there any evaluation of the fail reads for contamination QC?

1.11 The Paragraph starting on line 208 that describes the analysis of molecules transcribed with N1-methylpseudouridine could use some elaboration. How was the N1-methylpseudouridine base caller trained and tested (also lacking in the methods section) and what was the error rate with the canonical trained basecaller using the modified IVT molecules? I find this entire experimental section critical given that many mRNA vaccines are advertised using a complete replacement of canonical nucleotides with a modified form. Can the authors explain this section with more detail in the results and methods sections and expand on the results of this experiment in the discussion?

1.12 Line 95: the word “and” seems out of place in this sentence “This high level of contamination and was validated using Illumina...”

Response to reviewer comments: “Analysis of mRNA vaccines and therapies using VAX-seq”

We would like to thank the editors and reviewers for their time and consideration. Please find our responses to the reviewer's comments below that we agree have improved our manuscript.

Please note that in addition to our changes relating directly to reviewer comments, we would like to revise the manuscript title to “**mRNA vaccine quality analysis using RNA sequencing**”. We believe the new title reflects the evaluation of alternative RNA sequencing approaches (including short-, long-read sequencing and direct RNA) performed within the study, and will appeal to a broader audience.

Reviewer #1 (Remarks to the Author):

The manuscript by Gunter et al. seeks to address an important question in mRNA vaccinology: that is, how to study the complex mixture of RNA species that results from *in vitro* transcription of synthetic mRNA vaccines. This matters because double-stranded and other unintended RNA species can trigger innate immune sensing and inhibition of translation, not to mention issues for quality control and fragmented mRNA (during or after transcription). The unintended RNA species in mRNA vaccines are poorly characterized on the sequence level, and this area of research is likely to foster the development of superior (cheaper, faster, more reliable) methods of synthesizing mRNA vaccines. The authors use long-read nanopore sequencing as a promising technology to characterize these RNA species, comparing the data to short-read Illumina sequencing. As a disclaimer, my background is in mRNA vaccine design and immunology, and I do not have much background in the field of high-throughput sequencing. Therefore, I have focused my review on the value of this manuscript to the mRNA vaccine field.

The main strength that was demonstrated for the long-read sequencing method is the ability to detect truncations and other aberrant RNA species in a way that reveals the full length of each RNA species.

My enthusiasm for the value of this contribution was diminished by the fact that it reveals little specific information about the actual RNA sequences present in the mRNA vaccine samples, and a number of other scientific concerns detailed below.

Scientific concerns:

1.) Many of the figures show sequence coverage/alignment but lack specific information about the identity of the RNA species in question, in particular the unintended RNAs, which would be much more useful. Fig. 4e begins to touch on this, but the information presented here could be greatly expanded and made more quantitative and specific. In order to translate these results to the improvement of RNA vaccine design, we would need to see the actual aberrant RNA species that are present (sequence, length, etc). Where are the cryptic transcription start sites? Where is antisense transcription initiating from? What dsRNA species could potentially form? Are dsRNA species directly detectable using this method? All this will lead to models about how these aberrant RNA species are generated and how to avoid them if necessary.

We agree with the reviewer that the ONT long-read sequencing enables the detailed analysis of unintended RNAs that are spuriously generated during *in vitro* transcription. Therefore, we included an additional analysis of the off-target mRNAs architecture.

To analyse cryptic promoters, we analysed the start of each sequenced read to identify transcription start sites. This analysis identified at least one high-confidence cryptic promoter, with many full-length sequences initiating from the same site. These may have been the result of spurious T7 RNA polymerase binding. To describe this analysis we included the following text:

“The remaining (7.3%) of RNA species comprised different off-target RNAs. Of these, 0.3% likely initiated from a cryptic transcription start sites (Figure S12a-c).”

It is not possible to directly detect double-stranded RNAs using NGS, however, given the ONT direct RNA sequencing library preparation protocol preserves strandedness, we can infer dsRNA from the presence of anti-sense reads in

our datasets (however, this method is not sensitive to double-stranded RNA hairpins). This analysis is included in the Results section, under “Characterising off-target RNA contaminants”.

We compare these results to dsRNA immunoblotting (Sup Fig. S8e), and show that our methods are more sensitive than the current industry standard techniques.

In addition, many figures contain elements (in particular black, gray, white, and green colors) that are not defined and render it inaccessible to readers, especially those who are not deep in the field of high-throughput sequencing.

Thank you for your comment. We have simplified the IGV plots, and have now defined the colour coding in the figure legends and as keys within the figures.

2.) What are the details of the COVID-19 mRNA vaccines described here? Pfizer/BioNTech vs. Moderna? How were these obtained and treated? This information is necessary for these parts of the paper to be interpretable.

The COVID-19 mRNA vaccines described in this paper are neither the Pfizer nor Moderna COVID-19 vaccines, however they do encode sequences from the SARS-CoV-2 genome. The COVID-19 mRNA vaccine sequences are undergoing clinical development and their sequences are confidential and unfortunately cannot be disclosed. The COVID-19 mRNA vaccines were obtained and treated as for other mRNA vaccines manufactured at the BASE facility, and as indicated in the **Materials and Methods** section, and the **Fig. S4** legend.

3.) Can we distinguish the source of errors in mRNA sequence? (T7 transcription errors vs plasmid-derived errors vs hydrolysis after transcription, etc.)

By comparing the sequencing of plasmid DNAs relative to the mRNA, VAX-seq can distinguish between errors arising in mRNA *in vitro* transcription and errors arising during the preparation of plasmid templates, respectively. The following text has been included for clarification:

“A per-nucleotide comparison of the error profile between cDNA sequencing and previous plasmid DNA sequencing also reveals errors specific to the cDNA library preparation steps (Figure S6a).”

Additional errors such as abortive transcription, degradation and hydrolysis can also be inferred by analysing truncation patterns. For example, (i) abortive transcription can be inferred from 3' truncated reads, (ii) degradation by RNases can be inferred from 5' truncated reads and (iii) hydrolysis can be inferred from simultaneous 3' and 5' truncation. We have included this discussion in the manuscript as follows:

“For the eGFP mRNA, we found 58.2% of sequenced reads encompassed the full mRNA vaccine length (including the Kozak sequence, coding sequence and 3' UTR). Of the fragmented reads, 7% of reads were 3' truncated likely due to abortive transcription. The majority of the remaining sequences (31.3%) were 5' truncated likely due to degradation by RNases. The remaining 3.5% of reads displayed both 3' and 5' truncation and likely resulted from RNA hydrolysis.”

4.) I am concerned about the detection of a 66 nt deletion in the poly(A) tail of one of the COVID-19 vaccines using the tailfindr method, while no truncation was detectable by capillary or agarose electrophoresis (line 126). This appears to challenge the validity of their method.

We disagree. Capillary and agarose electrophoresis lack the sensitivity to detect small deletions. For example, Agilent list the sizing accuracy their D5000 (DNA) ScreenTape to be within +/- 10%. Therefore, a 66nt deletion should not be reliably detected in a ~700 nt fragment). By contrast, long-read sequencing of mRNA is sufficiently sensitive to detect a 66nt deletion in the difficult poly(A) tail. We also believe this deletion is a true positive because it was detected in both the plasmid and mRNA sequencing.

We have added the following text to clarify:

“This deletion was not distinguished by capillary electrophoresis, as this method has difficulty detecting small deletions (less than 10% of the mRNA fragment size).”

5.) What explains the different rates of identification of upstream, downstream, and antisense sequences between long read and short read methods, or is this discrepancy simply stochastic?

The different rates of identification of upstream, downstream, and antisense sequences between long- and short read sequencing methods is likely the result of both stochastic and systematic errors. For example, the uneven and heterogenous fragment coverage of short-read sequencing is highly reproducible between replicates.

“Short-read sequencing showed poor and uneven coverage that precluded an analysis of mRNA length and integrity (Figure 3a). This heterogenous alignment coverage was highly reproducible between replicates ($R^2=0.99$, Figure S9a,b).”

Another example of systematic sources of variation includes the 5' bias in short-read sequencing coverage that may partly explain the differential detection of aberrant 3' and 5' sequences compared to long-read sequencing, which shows a more even coverage. This 5' bias in short read cover is associated with a higher detection of fragments with 5' elongations, compared to long read sequencing (1.4% vs 0.3% respectively). Additionally, we observe an apparent lower abundance of (3') read-through transcripts in short-read vs long-read sequencing (5.7% vs 6.7% respectively).

No additional changes have been made to the main text.

6.) Fig. 5 shows a large fraction of truncated mRNAs identified by long read sequencing, yet this is not visible in the capillary electrophoresis shown in Fig. S11a, appearing to contradict what the authors state on line 206. If the nanopore sequencing does not agree with electrophoresis, does that mean there can be fragmentation artifacts in nanopore sequencing (hydrolysis, shearing, aborted sequencing, etc?), and might this be more likely with N1-methylpseudouridine?

A disadvantage of electrophoresis is that the method is not sensitive to mRNAs that have a wide range of fragment sizes (that often appear as a 'smear'). We also analysed the fragmented read profile depicted in Figure S11a that was generated through direct RNA sequencing. They predominantly included reads that were 5' truncated and may have been impacted by biases arising during library preparation. A lower proportion of 5' truncated reads was observed in our cDNA libraries that were generated from the same samples. To simplify this section, we have made the following edits:

“Transcript size distribution was also compared using capillary electrophoresis, for vaccines incorporating N1-methylpseudouridine and unmodified bases, with minor fragment size differences observed between the modified and unmodified vaccines (Figure S11a,b).”

7.) Given the error rate of 23.6% reported for N1-methylpseudouridine (line 213), it is unclear how robust the base-calling error method is for identifying this modified nucleoside. Also, if the error rate is higher for N1-methylpseudouridine compared to other nucleosides, it is unclear whether this is due to an inaccuracy of sequencing or perhaps an error rate in T7 transcription.

Currently, the Oxford Nanopore machine learning base-caller does not reliably identify modified nucleotides and makes errors at N1-methylpseudouridine. However, the Oxford Nanopore base-caller can be re-trained to detect modified nucleotides, such as N1-methylpseudouridine. Whilst this is possible (and has been performed for a number of native nucleotide modifications), this is outside the scope of this paper. However, the following text has been added to better explain:

“The consistent error profile at N1-methylpseudouridine suggests that retraining the base-caller will permit accurate detection modified nucleotides¹⁵.”

Other concerns:

1.) What is the positive control for dsRNA in the dot blot?

The positive control is a pathogen-specific double-stranded RNA (dsRNA) for virus resistance in plants that was generated at the BASE facility for use in agricultural applications (Mitter et. al., Nature Plants, 2017). This has been included in the figure legend as follows:

“dsRNA produced in-house, using sequences from Mitter et. al., Nature Plants, 2017 was blotted as a positive control and nuclease-free water for the negative control.”

2.) What is a “pre-nucleotide comparison” (line 208)?

Sorry, this is an error and has been corrected to “per-nucleotide comparison”.

3.) It is unclear which authors have the competing interest from Oxford Nanopore Technologies as there are no middle initials given in the author list and in some cases the authors have the same first and last initials.

Relevant middle initials have now been added to the author listing.

4.) Figure 2 – is this all for the eGFP mRNA? What is the negative polyadenylated control?

All of the main figures use the reference eGFP mRNA and plasmid sequences. The negative polyadenylated control is an eGFP mRNA that has a poly(A) tail added and extended using an *E. coli* poly(A) polymerase (see Methods and Materials). This has been clarified in the figure legend as follows:

“poly(A) tail length measured using *tailfindr* for eGFP mRNA (green), compared to a negative control eGFP mRNA with a variable poly(A) tail length, added using *E. coli* poly(A) polymerase.”

Reviewer #3 (Remarks to the Author):

The authors of “Analysis of mRNA vaccines and therapies using VAX-seq” used Nanopore sequencing to evaluate RNA samples at various steps during the manufacturing process to assess the Quality of the samples. They demonstrated that long reads were valuable for determining if the linearization process was completed, and if the IVT synthesis from those plasmids was successful. Gunter and Idrisoglu et al. found that Nanopore sequencing was useful for detecting IVT fragments and that both short-read and Nanopore sequencing were beneficial for detecting IVT synthesis contaminations, especially low abundance antisense products. The authors also evaluated short-read and Nanopore cDNA sequencing, and direct RNA nanopore sequencing on IVT synthesis products with N1-methylpseduouridine in place of canonical uridine, and found that although possible, direct RNA nanopore had much lower accuracy on these products.

This manuscript is well written, the experiments executed well, and the authors claims are supported by the evidence presented. Although it has not been previously demonstrated in the literature, this body of work represents an obvious but useful application of Nanopore sequencing. One thing that would help this manuscript is a description of how sensitive the sequencing needs to be to separate a good from a bad sample. For example, a sample that would pass QC by the orthogonal methods but fail when evaluated by sequencing would clearly demonstrate the advantage of using sequencing over the other established methods for mRNA vaccine QC.

We agree that demonstrating the advantage of long-read sequencing to detect manufacturing errors that are otherwise undetected using current quality control measures is key for adoption of this method. We highlight two examples in the manuscript where ONT sequencing detects errors undetected using methods currently proposed by the USP;

1. We detect antisense RNA (which indicates the presence of dsRNA) using both short read and long read sequencing, for samples where immunoblotting (the USP standard for dsRNA detection) does not detect any dsRNAs.
2. We detected a deletion in the poly(A) tail of a COVID-19 vaccine candidate through long read sequencing, that was otherwise undetected using capillary electrophoresis (the USP standard for RNA integrity measurement).

To highlight these examples, we include the following sentence in the discussion:

“For example, we detected dsRNAs using VAX-seq, which were not detectable through immunoblotting (the USP standard), and a partial deletion of the poly(A) tail using VAX-seq that was not detected with capillary electrophoresis.”

Below are some specific points for the authors:

1.1 Adding citations about the safety of mRNA vaccines would improve the first paragraph of the introduction.

Agreed, and we have now added citations on mRNA vaccine safety.

1.2 What did the other 2.2% of the reads align to if not the production strain genome when the antigenic sequences were used in the construct instead of the eGFP sequence. Also, do the authors know why there was substantially more contamination for the eGFP coding sequence than the antigenic sequences?

This has now been clarified in the text:

“In contrast, we did not detect any *E. coli* contamination in the COVID-19 mRNA vaccines, with 96.5% of plasmid reads aligning to the reference and 99.9% of mRNA reads aligning to the reference for our production vaccine (Figure S4a-c).”

The COVID-19 vaccines showed a substantial reduction in contamination of the plasmid preparations in comparison to the eGFP control mRNA vaccine, as improvements were made to the plasmid production SOPs during the preparation of these results.

1.3 How did the authors detect circular plasmid sequences with nanopore sequencing? The plasmid must be linearized (such as by the restriction enzyme, sonication, or the rapid library kit) before it can be sequenced by nanopore and this was not clear.

We agree that circularised plasmids are not detectable through ligation-based library preparation methods. However, it is possible to infer incomplete plasmid circularisation in mRNA vaccine sequencing through the presence of sequence that aligned to the plasmid sequence 3' to the *BsaI* recognition site. The following has been added to the text:

“Long-read nanopore sequencing can determine the full length of the linear plasmid, however ligation-based nanopore library preparation methods do not measure circular plasmids.”

1.4 Given that the IVT molecules are synthesized from 5' to 3' ends, I am surprised that such a high proportion of molecules (37%) were degraded from the 5' end. Was there a punctuated site where this degradation started/stopped, or was there a gradient? How many nucleotides were missing from the 5' end? If only 58.2% were full length, then why were 74% of the expected size? Does this mean that there was a contaminating nuclease that could bypass the m7G capped 5' end in the mRNA manufacturing process? Was this high rate of 5' end degradation also observed with the antigenic mRNA sequences? Can the authors elaborate further on this observation?

We defined “full length” sequences as sequences that include the full kozak sequence, coding sequence and 3' UTR, while “expected size” fragments were within 5% of the expected length. The discrepancy between the figures is due to counting fragments that were within 5% of the expected size (however did not include the full kozak sequence, coding sequence or 3' UTR). The following text was added to clarify:

“For the eGFP mRNA, we found 58.2% of sequenced reads encompassed the full mRNA vaccine length (including the full Kozak sequence, coding sequence and 3' UTR). Of the fragmented reads, 7% of reads were 3' truncated likely due to abortive transcription. The majority of the remaining sequences (31.3%) were 5' truncated likely due to degradation by RNases. The remaining 3.5% of reads displayed both 3' and 5' truncation and likely resulted from RNA hydrolysis.”

1.5 Figure 1, there is no (a) marker in the figure and referencing one in the legend seems unnecessary.

Thanks - this has now been deleted.

1.6 Figure 3, panel C is mislabeled as D in the figure legend.

Thanks - this has now been corrected.

1.7 Figure 4, a reference to panel A is missing.

Thanks - this has now been deleted.

1.8 When discussing the direct RNA nanopore sequencing error rate in the paragraph starting on line 183, was the error rate evaluated with and without including the poly(A) tail region in the reference? Much of the poly(A) tail should not have been basecalled by guppy and that may have increased the deletion error rate? It is not entirely clear from this paragraph how that was handled.

The analysis was performed without the poly(A) tail (which has a distinct error profile). The following changes have now been made:

“We first analysed mRNA vaccine quality using direct RNA sequencing, excluding the poly(A) tail.”

1.9 I couldn't find either of the two SRA accession IDs in the data accessibility section. To replicate the authors results, it is important to ensure that both the fastq and the fast5 files from the nanopore datasets are available for download because TailfindR was used.

FAST5 files have now been uploaded to SRA, for datasets that were analysed with *tailfindr*. The data will be made public prior to publication of our manuscript. We can organise data availability to the reviewer upon request.

1.10 Why was a quality cut off 9 used instead of 7, which is more typical of Nanopore experiments? Is it possible that more off-target reads can be found in the lower quality reads? Was there any evaluation of the fail reads for contamination QC?

We used a quality cut-off of 9 as this is the current standard cutoff for Guppy for high accuracy basecalling on the GridION instrument (earlier version of Guppy used a cut-off of 7).

1.11 The Paragraph starting on line 208 that describes the analysis of molecules transcribed with N1-methylpseudouridine could use some elaboration. How was the N1-methylpseudouridine base caller trained and tested (also lacking in the methods section) and what was the error rate with the canonical trained basecaller using the modified IVT molecules? I find this entire experimental section critical given that many mRNA vaccines are advertised using a complete replacement of canonical nucleotides with a modified form. Can the authors explain this section with more detail in the results and methods sections and expand on the results of this experiment in the discussion?

We did not train the base-caller to recognise N1-methylpseudouridine, but given the consistency of the error profiles observed, we anticipate that training the base-caller would improve the sensitivity for detecting modified nucleotides. Accordingly, the following sentence has been added:

“The consistent error profile at N1-methylpseudouridine suggests that retraining the base-caller will permit accurate detection modified nucleotides¹⁵.”

1.12 Line 95: the word “and” seems out of place in this sentence “This high level of contamination and was validated using Illumina...”

Thanks – this has now been deleted.

REVIEWER COMMENTS

Reviewer #1 (Remarks to the Author):

Below please find my new comments on the authors' responses, with the same numbering as my original comments about scientific and other concerns:

Scientific concerns:

1. I have additional questions regarding cryptic promoters/cryptic transcription start sites.

-Do you consider the presence of antisense transcripts to be evidence of cryptic transcription start sites?

-The legend of Fig S12 discusses T7 pol binding ("Most elongated transcripts display unique binding of T7 RNA polymerase") and affinity but I am not seeing any data for binding or affinity--hopefully I am not missing it. If you did not conduct direct assays of T7 binding, please rephrase in a more precise/cautious way. One alternative hypothesis to explain the 5' ends of the RNAs is hydrolysis, which could potentially be affected by secondary structures.

Re: dsRNA response: I am not sure I would call your method more sensitive since you haven't reported your threshold of detection. Also please define what "antibody 1" and "antibody 2" are. The mAb identities (e.g. clone J2 from X source) are important to interpret this.

Re: graph labels: Graphs are still often unclear. Some graphs (e.g. Fig 2a) don't contain any of the colors you indicated for the 4 nucleobases and instead have black or purple stripes. The purple arrow heads, I-beams, and purple numbers are not explained in figure S12. In graphs where you label "mismatch errors" with the 4 nucleobases, it is unclear if the labeled bases refer to the original templated nucleobases or the mutated versions. Why does your antisense RNA diagram (Fig 4e) have a polyA site after the antisense 5' UTR? Is this correct or a mistake?

2. Re: Covid mRNA vaccines: I don't think we need the exact proprietary sequences to interpret the paper, but I think certain elements would be important for readers to know. What protein(s) or kind of protein(s) do they encode and how long is this coding sequence? I would at least want to know if these encode the spike protein or not so we can understand if this has any relation to the Covid vaccines currently in use.

3. Follow up: Fig S4 shows a truncation in the poly(A) in a plasmid prep but I could not find it discussed in the text. Please explain this and if it was used to make any of the sequenced RNA vaccine.

4. Re: 66 nt deletion: Is it possible to run an agarose gel or TapeStation under different conditions and/or with a greater amount of loaded material loaded to try to resolve the 66 nt deletion? Since your method is a relatively novel way to examine mRNA vaccines it is important to corroborate it using a well understood method to guard against artifacts.

Additionally, please provide details about the batches in the different lanes of Fig S6c. Are these all the Covid mRNA vaccine (not GFP?). Are they all made from the same or different plasmid preps?

5. Are the reads shown in the "Alignments" section of Fig 3a only the unique (de-duplicated) reads? If so, this would explain the seeming discrepancy between the apparent high abundance of reads aligned to the region 3' of the polyA sequence and the low level of "coverage" in that same region. Please specify in legends.

Additionally, please comment on whether these post-polyA sequences are likely deriving from plasmid carryover into the RNA sample that was directly sequenced or from RNA made from non-linearized plasmid.

6. Re: not detecting RNA fragments by electrophoresis: The explanation provided by the authors on this issue is not yet satisfactory. The fragment peaks in Fig. 5c are at roughly half the max amplitude of the main mRNA peak and have a large AUC, maybe 25-30% of the main mRNA peaks. And the width of the fragment peaks is not so much larger than the main RNA peaks. There are minor bands/peaks in the gel electrophoresis in fig. S11 that appear to be at or below this abundance level, for example at 300 bp, and this is detectable by electrophoresis. For these reasons, I would need to see experimental evidence to support the claim/assumption that fragments like those in Fig 5c would not be visible in the electrophoresis. Alternatively, this could be discussed in a manner that considers alternative explanations for the fragment appearance in Fig. 5c.

7. No comments.

"Other concerns" #1-4: No comments.

Reviewer #3 (Remarks to the Author):

Gunter and Idrisoglu et al. have taken my comments into consideration and made many improvements to their initial manuscript. The scope and goals have been more clearly defined and the results have been made clearer. The authors have addressed all my initial comments in a satisfactory way, and no new issues came about because of the rework. I especially thank the authors for making the fast5 files available in addition to the fastq files with SRA, as it is less common than it should be. However, I do not need the files in advance of the eventual public release of those data.

A very minor point, but in the 5' degraded diagram in figure 4e, I would not expect to see a 5' cap as part of an RNase digested transcript. The way it is depicted would suggest a cryptic T7 promoter rather than RNase digestion.

RESPONSE TO REVIEWER COMMENTS

“MRNA VACCINE QUALITY ANALYSIS USING RNA SEQUENCING” (NCOMMS-22-40899A)

We would like to thank the editors and reviewers for their time and consideration. Please find our responses to the reviewer's comments below.

We appreciate the concern (highlighted in Reviewer #1's comments) about providing further details on the COVID-19 mRNA vaccines analysed in our manuscript, and the release of associated sequencing data. We have asked whether we could publish this data, however, unfortunately, our partners are unable to release the COVID-19 mRNA vaccine sequence data due to commercial confidentiality including Intellectual Property considerations.

Given this, please let us know how you would like to proceed. Although we believe that the COVID-19 mRNA vaccine sequences and data are not necessary to demonstrate the use of RNA sequencing to analyse manufacturing failures. However, if necessary, we can remove Figure S4, and any text related to these sequences of COVID-19 mRNA vaccines. All other data described in this manuscript have been deposited in public databases and will be released at the time of publication.

Reviewer #1 (Remarks to the Author):

Below please find my new comments on the authors' responses, with the same numbering as my original comments about scientific and other concerns:

Scientific concerns:

1. I have additional questions regarding cryptic promoters/cryptic transcription start sites.

-Do you consider the presence of antisense transcripts to be evidence of cryptic transcription start sites?

Thank you for your comment. The presence of antisense and double-stranded transcripts may derive from cryptic promoters on the negative strand. However, previous studies (e.g. Mu et al. 2018) suggest these antisense RNAs are generated from aberrant priming and atypical transcription initiation from the mRNA 3' end. We have clarified this in the main text as follows:

“For comparison, we also analysed synthetic mRNAs with short-read (Illumina) cDNA sequencing. This method uses random hexamer priming and can sensitively detect non-polyadenylated RNA, such as truncated or antisense RNAs (Figure 4e, Figure S8b), which were likely derived from aberrant transcription, initiated 3' to the poly(A) tail.”

-The legend of Fig S12 discusses T7 pol binding ("Most elongated transcripts display unique binding of T7 RNA polymerase") and affinity but I am not seeing any data for binding or affinity--hopefully I am not missing it. If you did not conduct direct assays of T7 binding, please rephrase in a more precise/cautious way. One alternative hypothesis to explain the 5' ends of the RNAs is hydrolysis, which could potentially be affected by secondary structures.

We have not performed affinity assays to analyse the binding and transcription initiation of T7 RNA Polymerase to the plasmid DNA template. Whilst this sounds useful to better understand the role of cryptic transcription initiation in generating off-target RNA transcripts, this is beyond the scope of the study. Nevertheless, as suggested, we have rephrased the main text as follows:

“The impact of cryptic promoters, and their contribution to off-target RNA species can be further investigated by techniques such as DNA footprinting.”

Re: dsRNA response: I am not sure I would call your method more sensitive since you haven't reported your threshold of detection. Also please define what “antibody 1” and “antibody 2” are. The mAb identities (e.g. clone J2 from X source) are important to interpret this.

Thanks for your comment, and we have amended the Results section as follows:

“This method for the indirect detection of dsRNAs may provide an alternative to dsRNA immunoblotting, the current industry standard technique for dsRNA detection (**Figure S8e**)⁷.”

And the Discussion:

“VAX-seq was shown to provide a quantitative and sensitive measure of mRNA features, providing a streamlined alternative to the currently recommended analytical techniques. For example, current industry standard techniques include immunoblotting to detect dsRNA, and capillary electrophoresis and RP-HPLC for mRNA integrity analysis. VAX-seq provides both antisense RNA detection (an indirect indicator of dsRNA) and mRNA integrity analysis, alongside sequence confirmation, in a single step protocol that can be performed in real-time. This enables rapid testing within hours of manufacture so quality control issues can be quickly detected to inform trouble-shooting.”

Additionally, we have added details of the dsRNA-specific antibodies in the Methods section:

“For the detection of dsRNA, the membrane was incubated overnight at 4°C with two different dsRNA-specific murine monoclonal antibodies (mAb), that were derived from clones 3G1 and 2G4 (Mozzy Mabs, Brisbane, Australia). Both antibodies were incubated separately overnight at a 1:5 dilution, diluted in incubation buffer (1% Non-Fat Dry Milk in TBS-T).”

Additionally, the dot blots relating to each antibody are labelled with clone numbers in **Figure S8e**.

Re: graph labels: Graphs are still often unclear. Some graphs (e.g. Fig 2a) don't contain any of the colors you indicated for the 4 nucleobases and instead have black or purple stripes. The purple arrow heads, I-beams, and purple numbers are not explained in figure S12. In graphs where you label “mismatch errors” with the 4 nucleobases, it is unclear if the labeled bases refer to the original templated nucleobases or the mutated versions.

Thanks for your feedback. The purple and black lines in the figure indicate insertions and deletions respectively. This has been clarified in the **Figure 2a** key, as well as other relevant figures (such as **Figures S8** and **S12**).

Why does your antisense RNA diagram (Fig 4e) have a polyA site after the antisense 5' UTR? Is this correct or a mistake?

Thank you for your comment. We originally illustrated the antisense transcript as it requires an additional polyadenylation step conducted prior to library prep (which selects for polyadenylated transcripts). However, we agree that this illustration may be confusing, and therefore the poly(A) tail has been replaced by a 3' T homopolymer.

2. Re: Covid mRNA vaccines: I don't think we need the exact proprietary sequences to interpret the paper, but I think certain elements would be important for readers to know. What protein(s) or kind of protein(s) do they encode and how long is this coding sequence? I would at least want to know if these encode the spike protein or not so we can understand if this has any relation to the Covid vaccines currently in use.

The COVID-19 mRNA vaccines are ~3kb and encode the SARS-CoV-2 Spike protein (Wuhan). However, there have been some modifications to this antigen sequence that is slightly different to the mRNA-1273 (Moderna) and Comirnaty (BioNTech/Pfizer) mRNA vaccines.

As a result, the sequence has been considered confidential by our partners, and we are unable to publish the full sequence. However, for our purposes, the COVID-19 vaccines are solely used as an example to illustrate the use of RNA sequencing to analyse mRNA vaccines and identify manufacturing failures. The COVID-19 mRNA vaccines were obtained and treated as other mRNA vaccines manufactured at the BASE facility, as indicated in the **Materials and Methods** section, and the **Fig. S4** legend.

The following text has been added to the Methods section:

“The same methods were used for the manufacture of two experimental COVID-19 vaccines.”

And the Figure S4 legend:

“Plasmid sequencing of novel COVID-19 vaccine candidates manufactured at the BASE facility, aligned to their reference genomes.”

3. Follow up: Fig S4 shows a truncation in the poly(A) in a plasmid prep but I could not find it discussed in the text. Please explain this and if it was used to make any of the sequenced RNA vaccine.

Thank you for your comment. The following sentence has now been added to the Results section:

“This poly(A) tail deletion was also detected in the plasmid template (**Figure S4a**).”

4. Re: 66 nt deletion: Is it possible to run an agarose gel or TapeStation under different conditions and/or with a greater amount of loaded material loaded to try to resolve the 66 nt deletion? Since your method is a relatively novel way to examine mRNA vaccines it is important to corroborate it using a well understood method to guard against artifacts.

Resolving the impact of a 66nt deletion on the migration of a ~1,200nt mRNA is difficult and is within the estimated 10% size error for DNA Tapes reported by Agilent. Additionally, no size benchmarking has been reported by Agilent for RNA Tapes, so their precision in size analysis is unclear. Whilst we agree it may be possible to optimise the agarose gel electrophoresis conditions to resolve this deletion, it is not feasible to routinely perform such optimisation on individual mRNAs, and is beyond the scope of the manuscript.

Additionally, please provide details about the batches in the different lanes of Fig S6c. Are these all the Covid mRNA vaccine (not GFP?). Are they all made from the same or different plasmid preps?

Figure S6c shows replicate eGFP control mRNAs that have been prepared from a single plasmid preparation. This information has now been added to the **Figure S6c** legend:

“Capillary electrophoresis size profile of reference eGFP mRNA vaccines (as measured with *Agilent TapeStation*) shows size profile (left) and capillary electrophoresis bands for replicate eGFP mRNAs transcribed from single plasmid preparation (right).”

5. Are the reads shown in the "Alignments" section of Fig 3a only the unique (de-duplicated) reads? If so, this would explain the seeming discrepancy between the apparent high abundance of reads aligned to the region 3' of the polyA sequence and the low level of "coverage" in that same region. Please specify in legends.

The “Alignments” section of this plot includes non-unique alignments, collapsed into a single line by default in IGV. Additionally, the bottom part of the figure has been cropped to fit within the space constraints of the journal. This has now been clarified in the **Figure 3a** legend, and other relevant figure legends:

“Genome-browser (IGV) view of short-read cDNA sequenced libraries aligned to the reference plasmid sequence. Coverage indicates the alignment depth at each nucleotide position, whilst the lower alignments grey bars indicate unique, individual alignments, with colouring indicating their similarity to the reference genome.”

Additionally, please comment on whether these post-polyA sequences are likely deriving from plasmid carryover into the RNA sample that was directly sequenced or from RNA made from non-linearized plasmid.

The synthesised mRNA is treated with DNase and then purified to remove the residual DNA template. This removal is confirmed with TapeStation prior to library prep, however, it is possible that trace amounts of plasmid DNA molecules, below the detection limit of the TapeStation, may be included as carryover contamination in the RNA sequencing library. The following has been added to the Results section:

“Of these, 0.3% were likely derived from cryptic transcription start sites (Figure S12a-c) or derived from residual plasmid DNA template...”

6. Re: not detecting RNA fragments by electrophoresis: The explanation provided by the authors on this issue is not yet satisfactory. The fragment peaks in Fig. 5c are at roughly half the max amplitude of the main mRNA peak and have a large AUC, maybe 25-30% of the main mRNA peaks. And the width of the fragment peaks is not so much larger than the main RNA peaks. There are minor bands/peaks in the gel electrophoresis in fig. S11 that appear to be at or below this abundance level, for example at 300 bp, and this is detectable by electrophoresis. For these reasons, I would need to see experimental evidence to support the claim/assumption that fragments like those in Fig 5c would not be visible in the electrophoresis. Alternatively, this could be discussed in a manner that considers alternative explanations for the fragment appearance in Fig. 5c.

Thank you for your comment. We agree that the truncated fragments detected in our RNA sequencing are likely to present at lower levels in our capillary electrophoresis traces. However, because they have a variable size, they do not form a distinct peak or species, and are therefore refractory to analysis with electrophoresis. As requested by the reviewer, we discuss this in the main text and discussion as follows:

“This size distribution profile calculated from the read-length is analogous to measurement using electrophoretic methods (Agilent TapeStation; Figure S6c). However, an advantage of RNA sequencing is that individual peaks can be analysed to determine the mRNA sequences.”

7. No comments.

"Other concerns" #1-4: No comments.

N/A

Reviewer #3 (Remarks to the Author):

Gunter and Idrisoglu et al. have taken my comments into consideration and made many improvements to their initial manuscript. The scope and goals have been more clearly defined and the results have been made clearer. The authors have addressed all my initial comments in a satisfactory way, and no new issues came about because of the rework. I especially thank the authors for making the fast5 files available in addition to the fastq files with SRA, as it is less common than it should be. However, I do not need the files in advance of the eventual public release of those data. A very minor point, but in the 5' degraded diagram in figure 4e, I would not expect to see a 5' cap as part of an RNase digested transcript. The way it is depicted would suggest a cryptic T7 promoter rather than RNase digestion.

We appreciate the reviewer's feedback and effort, and improvements to the manuscript. This figure has now been amended as suggested.

REVIEWER COMMENTS

Reviewer #1 (Remarks to the Author):

Thank you for addressing several specific scientific concerns. Given the discussion of proprietary Covid sequence information, my overall impression is that the overall impact of the paper will be moderately interesting to people who quality control mRNA vaccine production but of lower utility to basic researchers trying to improve mRNA vaccine production without sequence-level information, which it seems is impossible for the Covid vaccines but potentially possible for the GFP mRNA.